# Cathode engineering with perylene-diimide interlayer enabling over 17% efficiency single-junction organic solar cells

Jia Yao[1], Beibei Qiu[2], Zhi-Guo Zhang [1✉], Lingwei Xue[1], Rui Wang[3], Chunfeng Zhang [3], Shanshan Chen[4,5], Qiuju Zhou[6], Chenkai Sun [2,7,8], Changduk Yang [4], Min Xiao [3], Lei Meng[2] & Yongfang Li[2,7✉]

In organic solar cells (OSCs), cathode interfacial materials are generally designed with highly polar groups to increase the capability of lowering the work function of cathode. However, the strong polar group could result in a high surface energy and poor physical contact at the active layer surface, posing a challenge for interlayer engineering to address the trade-off between device stability and efficiency. Herein, we report a hydrogen-bonding interfacial material, aliphatic amine-functionalized perylene-diimide (PDINN), which simultaneously down-shifts the work function of the air stable cathodes (silver and copper), and maintains good interfacial contact with the active layer. The OSCs based on PDINN engineered silver-cathode demonstrate a high power conversion efficiency of 17.23% (certified value 16.77% by NREL) and high stability. Our results indicate that PDINN is an effective cathode interfacial material and interlayer engineering via suitable intermolecular interactions is a feasible approach to improve device performance of OSCs.

[1] State Key Laboratory of Organic/Inorganic Composites, Beijing Advanced Innovation Center for Soft Matter Science and Engineering, Beijing University of Chemical Technology, Beijing 100029, China. [2] Beijing National Laboratory for Molecular Sciences, CAS Key Laboratory of Organic Solids, Institute of Chemistry, Chinese Academy of Sciences, Beijing 100190, China. [3] National Laboratory of Solid State Microstructures, School of Physics, and Collaborative Innovation Center of Advanced Microstructures, Nanjing University, Nanjing 210093, China. [4] Department of Energy Engineering, School of Energy and Chemical Engineering, Low Dimensional Carbon Materials Center, Ulsan National Institute of Science and Technology (UNIST), Ulsan 689-798, South Korea. [5] MOE Key Laboratory of Low-grade Energy Utilization Technologies and Systems, CQU-NUS Renewable Energy Materials & Devices Joint Laboratory, School of Energy & Power Engineering, Chongqing University, Chongqing 400044, China. [6] Analysis & Testing Center, Xinyang Normal University, Xinyang, Henan 464000, China. [7] School of Chemical Science, University of Chinese Academy of Sciences, Beijing 100049, China. [8] College of Chemistry and Molecular Engineering, Zhengzhou University, Henan 450001, China. ✉email: zgzhangwhu@iccas.ac.cn; liyf@iccas.ac.cn

Organic solar cells (OSCs) have emerged as a compelling energy technology because of their advantages of simple device structure, lightweight, capability to be fabricated into flexible devices via layer-by-layer solution processing[1,2]. Recently, tremendous efforts are focused on designing narrow bandgap non-fullerene acceptors[3–7] to overcome the intrinsic drawbacks of fullerene acceptors[8,9]. These efforts together with polymer donor development have promoted the power conversion efficiency (PCE) of the OSCs to a high level of 15–16% for single-junction devices[10–15], demonstrating a bright future for real application of the OSCs. Besides the development of solar light harvesters[7,16], interface engineering[17–23], especially cathode interlayer engineering[24–29], has also played an important role for advancing the OSCs toward commercialization. However, fewer studies have aimed at cathode interlayer design in those non-fullerene-based OSCs to prevent carrier recombination at cathode and improve device stability, even though such cathode interlayer have turned out to be equally important as the active layer photovoltaic materials for developing efficient and stable OSCs (refs. [24–29]).

It has been successfully demonstrated that some alcohol/water soluble organic cathode interlayer materials (CIMs) can provide an energy-level alignment at the electrode interface, as well as an interfacial dipole for an ohmic contact[19,30–35]. As a typical example, we reported such effective CIMs, PDIN bearing amino group and PDINO bearing amino N-oxide group on perylenediimide (PDI)[36]. Now, they are widely used by many research groups (examples are collected in Supplementary Table 2). However, for those interlayers, due to their limited ability in lowering the work function (WF) of the top electrodes; they usually work well with Al top electrode, and that is why only PDINO/Al cathode was used to design high-efficiency non-fullerene OSCs (refs. [10,37]), tandem OSCs (ref. [38]), and even perovskite solar cells[39]. While the OSCs with Al as cathode show poor stability due to the high reactivity of Al electrode in air. Another concern of using Al electrode is the forming adduct with the organic interlayer during the vacuum deposit process, which will react efficiently with molecular oxygen and water[40]. Attempts to improve the device stability by replacing Al with Ag or Cu for top cathode, commonly result in a higher WF of the cathode, thus a lower open-circuit voltage ($V_{OC}$) of the OSCs due to a low built-in potential across the device[26].

A solution is to design CIMs with highly polar groups to increase its dipole moment and thus to increase its ability in WF tunability. However, this approach is constrained by the intrinsic high surface tension of the CIMs bearing the polar group, which harms its solution deposition on active layer, and could result in poor contact between the CIM and the active layer[26,30–34]. It is well-known that the physically poor contact hampers electron collection by the electrode, which is one of the reasons for the low fill factor (FF) in OSCs (ref. [41]). In addition, the decohesion of the weak interface between the CIM and the active layer is inevitable for long-time device operation, and this is more severe in large area devices. In organic electronics, such as organic light-emitting diode, such interfacial issue is considered as one of the origin of the device deterioration[42]. Thus, it remains challenging to design CIMs that possess suitable dipole moment for decreasing WF of the air stable metals (such as Ag), and simultaneously, provide good interfacial compatibility with the active layer below and suppress carrier recombination and interlayer decohesion.

With these concerns, in this study, we develop an aliphatic amine group functionalized PDI derivative, namely PDINN, for the application as CIM in the OSCs with air stable metals Ag and Cu as top cathode. PDINN possesses suitable dipole moment to decrease WF of air stable metals Ag and Cu and the secondary amine in the side chains of PDINN can form hydrogen bonding with the photovoltaic materials in the active layer that results in good contact with active layer. In addition, PDINN can be facilely synthesized by a typical one-step approach under a large scale of over 64 g in lab from cheap raw materials, which is different from most CIMs that can only be accessible on milligram scale due to inevitable verbose multisteps synthesis and tedious purification steps. Using PDINN as cathode interlayer and Ag as the top cathode, OSCs with PM6 as donor and Y6 as acceptor[10] (molecular structures are shown in Supplementary Fig. 1) exhibit a high PCE of 17.23% (certified PCE of 16.77% by National Renewable Energy Laboratory (NREL)). The results indicate that PDINN is a high-performance and low-cost CIM for future industrial application of the OSCs.

## Results

**Materials synthesis and characterization.** The device structure is provided in Fig. 1a, and the photovoltaic materials (PM6 and Y6) for the active layer are shown in Supplementary Fig. 1. As shown in Fig. 1b, PDINN was easily prepared in one step, by condensation of N,N-dimethyldipropylenetriamine with perylene-3,4,9,10-tetracarboxylic dianhydride in methanol. The chemical structure of PDINN was verified by mass spectrometry (Supplementary Fig. 2) and NMR spectroscopy (Supplementary Fig. 3). The reaction proceeded almost entirely in methanol with a yield of ca. 95%. Notably, 64.4 g large-scale syntheses of PDINN can be readily conducted in methanol under mild reaction conditions (70 °C) and short reaction time (8 h). And the condensation reaction is convenient for further scale-up if needed. With the model for cost calculation[43], the cost of material ($C_g$, cost-per-gram) for PDINN is 1.6 \$ $g^{-1}$ (Supplementary Table 1). The $C_g$ value is less than one fifth of the well-known low-cost donor P3HT (10.0 \$ $g^{-1}$), and less than one percent of the fullerene (240 \$ $g^{-1}$), which is the starting material to synthesize effective fullerene derivative interlayers[25]. The result indicates that PDINN is really a low-cost material convenient for large-scale application.

To better understand the role of the aliphatic amine group, the thermal, physicochemical, and the photovoltaic properties of PDINN are compared with the widely used PDINO CIM. The thermal stability of PDINN and PDINO was investigated by putting the two samples on a hot plate under 150 °C for 1 h followed by dissolving the samples in methanol. It can be seen that PDINO becomes insoluble in methanol, whereas, clear PDINN solution can be still obtained for the PDINN sample after the thermal treatment (Supplementary Fig. 4). The result indicates that PDINN with the aliphatic amine group is more stable than PDINO with N-oxide amine group. The good thermal stability of PDINN is further confirmed by their thermogravimetric analysis (TGA), which show the 5% weight loss temperature at 103.4 °C for PDINIO and 240.1 °C for PDINN (Supplementary Fig. 4b).

**Intermolecular interactions with active layer.** PDINN shows a good solubility of 26.7 mg $mL^{-1}$ in methanol without the assistance of any acid, while most organic CIMs (such as PDIN or PFN) need hydrochloric acid or acetic acid to make a clear alcohol solution for processing[24,36]. In addition, PDINN shows a better interfacial compatibility on a wide range of active layers of OSCs compared with PDINO. For example, when a dilute solution (0.5 mg $mL^{-1}$) of PDINN or PDINO was spin-coated onto the PM6:Y6 active layer at high spin speed (5000 rpm), PDINN can form a uniform film, whereas PDINO shows a poor film formation, as shown in the photo-induced force microscopy (PiFM) images in Supplementary Fig. 5. This phenomenon is likely related to the hydrogen bond formed between the secondary amine in the aliphatic amine groups of PDINN with the active layer surface.

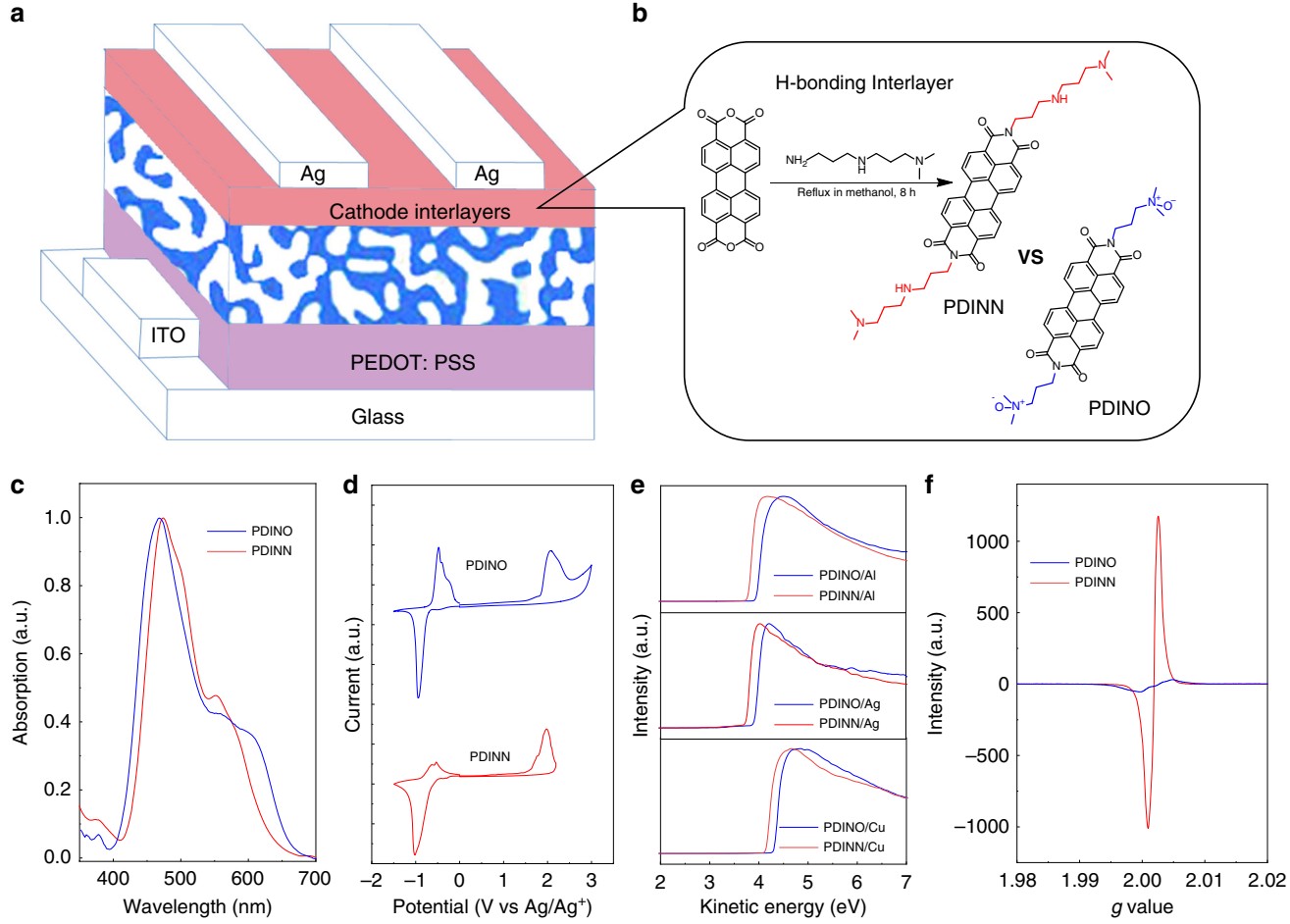

**Fig. 1 Chemical structures and characterization of the PDI derivatives CIMs. a** Device structure of the OSCs. **b** Synthetic route of PDINN along with the chemical structures of PDINN and PDINO. **c** Normalized UV–vis film absorption spectra of PDINN and PDINO films. **d** Cyclic voltammograms of PDINN and PDINO. **e** UPS spectra of the PDINN and PDINO-treated electrodes. **f** Electron spin resonance spectra of PDINN and PDINO in solid.

To confirm this, we initially conducted the variable-temperature and titration $^1$H NMR experiments, with PDINN and Y6 in solution (Supplementary Fig. 6). The $^1$H NMR signal at δ1.54 (assigned to hydrogen in the secondary amine) broadens and downfield shifts. To provide more evidence on the interactions between Y6 in the active layer and PDINN CIM in the OSC devices, we performed attenuated total reflectance (ATR) FT-IR measurement for the solid blend sample of Y6 and PDINN (w/w, 1:1), and the result is shown in Supplementary Fig. 7a. In contrast to the ATR FT-IR spectrum of pure Y6 sample with the C–F mean peak at 1350 cm$^{-1}$, the C–F peak of the Y6/PDINN blend sample moves to 1297 cm$^{-1}$ and is broadened. Moreover, we further studied the interaction between PDINN and solid Y6 by measuring $^{19}$F NMR spectra of solid Y6 power in methanol-D$_4$ NMR tubes with and without PDINN. The $^{19}$F NMR spectra of the samples are shown in Supplementary Fig. 7b. It can be seen that there is a $^{19}$F signal at −152 ppm in the PDINN methanol-D$_4$ solution, which is from the F atom of Y6. However, for the control sample without PDINN in methanol-D$_4$, there is no such signal for Y6 because Y6 is not soluble in methanol. Thus, the above results collectively provide strong evidence for the formation of the hydrogen bond between secondary amine group of PDINN and C–F group of Y6 in the solid state. The results also suggest that PDINN can provide intermolecular interaction with the organic semiconductor (especially those bearing fluorine groups[4]), such as Y6 during the deposition of PDINN from its methanol solution. The hydrogen-bonding feature of PDINN

indicates that PDINN as CIM in OSCs can improve the interfacial compatibility, with the active layer blow via a spontaneous adsorption process driven by the hydrogen-bonding interactions, thus reduce the series resistance of the devices[44].

**Physicochemical properties**. Figure 1c shows UV–vis absorption spectra of PDINN and PDINO film. PDINN displays a blue-shifted absorption relative to that of PDINO, with an absorption edge at 630 nm and a maximum absorption at 473 nm. To explain the phenomenon of the blue-shifted absorption, we measured the hydrogen-bonding property of PDINN in comparison with PDIN (ref. [36]) by measuring the ATR FT-IR spectra of PDINN and PDIN, as shown in Supplementary Fig. 7c. The carbonyl vibration at 1692 cm$^{-1}$ for PDIN is shifted to 1687 cm$^{-1}$ for PDINN, indicating the hydrogen-bonding formation in PDINN. Therefore, the blue-shifted absorption of PDINN film could be related to the different aggregation of PDINN molecules in comparison with the PDINO molecules due to its hydrogen-bonding property[45]. Cyclic voltammetry measurements were performed to measure the highest occupied molecular orbital (HOMO) and lowest unoccupied molecular orbital (LUMO) energy levels ($E_{HOMO/LUMO}$) of the materials. The onset oxidation/reduction potentials ($\varphi_{ox/red}$) can be obtained from the cyclic voltammograms, as shown in Fig 1d. According to the equations of $E_{HOMO/LUMO} = -e\ (\varphi_{ox/red} - \varphi_{Fc^+/Fc} + 4.8)$ (eV), the $E_{LUMO}$ value was calculated to be −3.78 eV for PDINN and −3.63 eV for

**Table 1 Comparison of the physicochemical properties of the CIMs of PDINO and PDINN.**

| Interlayer | Film absorption (nm) | | HOMO (eV) | LUMO (eV) | Conductivity ($10^{-5}$ S cm$^{-1}$) | Work function (eV)[a] | | |
|---|---|---|---|---|---|---|---|---|
| | $\lambda_{max}$ | $\lambda_{edg}$ | | | | Al | Ag | Cu |
| PDINO | 468 | 675 | −6.21 | −3.63 | 0.24 | 3.83 | 3.88 | 4.24 |
| PDINN | 473 | 630 | −6.02 | −3.78 | 50 | 3.67 | 3.72 | 4.08 |

[a]The work function of the metal cathode modified by the cathode interlayer material.

PDINO. The LUMO energy-level values of the PDI derivatives, especially for PDINN, are close to those of the non-fullerene acceptors[4,9], which can form a better energy-level alignment at the cathode interface and hence enhance electron collections. And the low-lying $E_{HOMO}$ values of −6.02 eV for PDINN and −6.21 eV for PDINO mean that holes from various donors will be blocked sufficiently at the cathodes, with the PDI derivative CIMs. Grazing-incidence wide-angle X-ray scattering (GIWAXS) was used to measure the crystallinity of the PDI derivatives deposited on silicon substrates. As shown in Supplementary Fig. 8, both of the two PDI interlayers show a semicrystalline structure, where their defined peaks are associated with their sufficient aggregation of the PDI core in the solid for efficient charge transport, when they are used as the CIMs. With a smaller terminal group in PDINO, although it shows a stronger aggregation behavior, its predominant edge-on orientation as revealed by the strong (100) peak in the out-of-plane direction may weaken its carrier extraction ability from the active layer.

Ultraviolet photoelectron spectroscopy (UPS) was used to probe the WF values of different metal electrodes of Al, Ag, and Cu modified with the CIMs. The WF modification ability is frequently associated with the interfacial dipole provided by the CIMs (refs. [19,26,32,46,47]). From Fig. 1e and Table 1, it can be seen that the two CIMs successfully reduced the WFs of the electrodes, the WF values of Al, Ag, and Cu were reduced to 3.83, 3.88, and 4.24 eV for the PDINO CIM, and 3.67, 3.72, and 4.08 eV for the PDINN CIM, respectively (see Table 1). PDINN shows a stronger WF lowering ability over that of PDINO for each electrode, with ca. 0.16 eV lower WFs for the PDINN-modified electrodes than that of the PDINO-modified electrodes. It is well accepted that the dipole moment in the amine group-containing CIMs is originated from the electron transfer from the nitrogen lone pair of neutral amine group to the electrode[20,26]. Thereby, the lower WFs obtained for PDINN should be associated with its higher dipole moment from the aliphatic amine group, which is just as poly(ethyleneimine) (PEI) or its derivative does in effectively lowering the WFs of different electrodes[20,48]. These results encouraged us to explore PDINN as cathode interlayer with high WF metals (such as Ag and Cu) as cathode.

The doping effects of the CIMs were investigated by the electron spin resonance (ESR) spectroscopy (Fig. 1f). PDINN shows a strong self-doping behavior (g value of 2.001) relative to PDINO (g value at 2.002), which is enabled by the unpaired electrons in aliphatic amine group of PDINN and beneficial to improving the electrical conductivity of the materials[46,49,50]. The electrical conductivities of the CIM films were measured by depositing two parallel silver electrodes on the two sides of the films. The conductivities of PDINO and PDINN calculated from the I–V curves (Supplementary Fig. 9) are $2.4 \times 10^{-6}$ S cm$^{-1}$ and $5.0 \times 10^{-4}$ S cm$^{-1}$, respectively, which indicate that PDINN has a better electron-transport ability.

**Photovoltaic performance**. To evaluate the photovoltaic performance of the PDINN CIM in OSCs, we fabricated conventional

**Table 2 Photovoltaic performance of the OSCs base on PM6:Y6 with Ag as cathode under the illumination of AM 1.5 G, 100 mW cm$^{-2}$.**

| Devices | $V_{oc}$ (V) | $J_{sc}$ (mA cm$^{-2}$) | FF (%) | PCE (%)[b] |
|---|---|---|---|---|
| w/o | 0.787 | 24.98 (24.86[a]) | 70.40 | 13.84 |
| | 0.784 ± 0.002 | 24.57 ± 0.36 | 69.51 ± 1.81 | 13.39 ± 0.53 |
| PDINO | 0.821 | 25.58 (24.94[a]) | 72.24 | 15.17 |
| | 0.816 ± 0.003 | 25.47 ± 0.25 | 71.80 ± 1.85 | 14.94 ± 0.42 |
| PDINN | 0.847 | 25.89 (25.76[a]) | 78.59 | 17.23 |
| | 0.845 ± 0.004 | 25.51 ± 0.28 | 77.84 ± 0.80 | 16.78 ± 0.33 |
| PDINN[c] | 0.843 | 25.704 | 77.5 | 16.77 |

[a]Calculated $J_{sc}$ from IPCE.
[b]Average values with standard deviations were obtained from ten devices.
[c]Certified result from National Renewable Energy Laboratory (NREL), USA.

devices with a structure of indium tin oxide (ITO)/PEDOT: PSS (poly (3,4-ethylenedioxythiophene): poly(styrene-sulfonate)) /PM6:Y6/CIM/electrode. As for the active layer, PM6: Y6 blend represents the state-of-the-art system with initially reported efficiency of 15.7% by Zou, using the PDINO/Al cathode[10]. Recently, intensive device engineering with adding a third component in the PM6:Y6 system increased the efficiency of the ternary OSCs over 16.5% (refs. [51–53]). Without tedious component optimization, here, we demonstrate a simple yet effective cathode modification with PDINN to realize a high efficiency over 17% for binary PM6:Y6 OSC (see Table 2). To better understand the high performance of PDINN interlayer, the device performance of PDINN was compared with that of PDINO, especially for the devices with air stable metal Ag and Cu as cathode.

Current density–voltage (J–V) characteristics of the optimal OSCs without the CIM (treated with methanol), and those with the CIM are compared in Fig. 2a, and the relevant photovoltaic parameters are listed in Table 2. Figure 2b shows the incident photon-to-converted current efficiency (IPCE) curves, and the integrated currents from the IPCE spectra agree quite well with the $J_{sc}$ values measured from the J–V curves. The PDINO/Ag device delivers a good PCE of 15.17%, along with a $V_{oc}$ of 0.821 V, a $J_{sc}$ of 25.58 mA cm$^{-2}$, and an FF of 72.24%, which is much higher than that (13.84%) of the Ag-only device. The OSC devices with PDINN/Ag cathode demonstrated an even higher maximum PCE of 17.23%, the remarkable improvement in PCE of the PDINN/Ag-based device stems from the substantial increase in the key device parameters of $V_{oc}$ (from 0.821 V to 0.847 V) and FF (from 72.24 to 78.59%) in comparison with the PDINO/Ag-based device. The efficiency histograms of PDINN-based and PDINO-based devices are provided in Supplementary Fig. 10. The photovoltaic performance of the optimized PDINN/Ag device was certified by NREL, and a PCE of 16.77% together with a $V_{oc}$ of 0.843 V, an FF of 77.5%, and a $J_{SC}$ of 25.704 mA cm$^{-2}$ was confirmed (Supplementary Fig. 11).

Interestingly, the photovoltaic performance of the PDINN-based OSCs exhibit good tolerance to the PDINN thickness variation in fabricating the OSC devices, which is important for large area fabrication of the OSCs. While the commonly used PEI

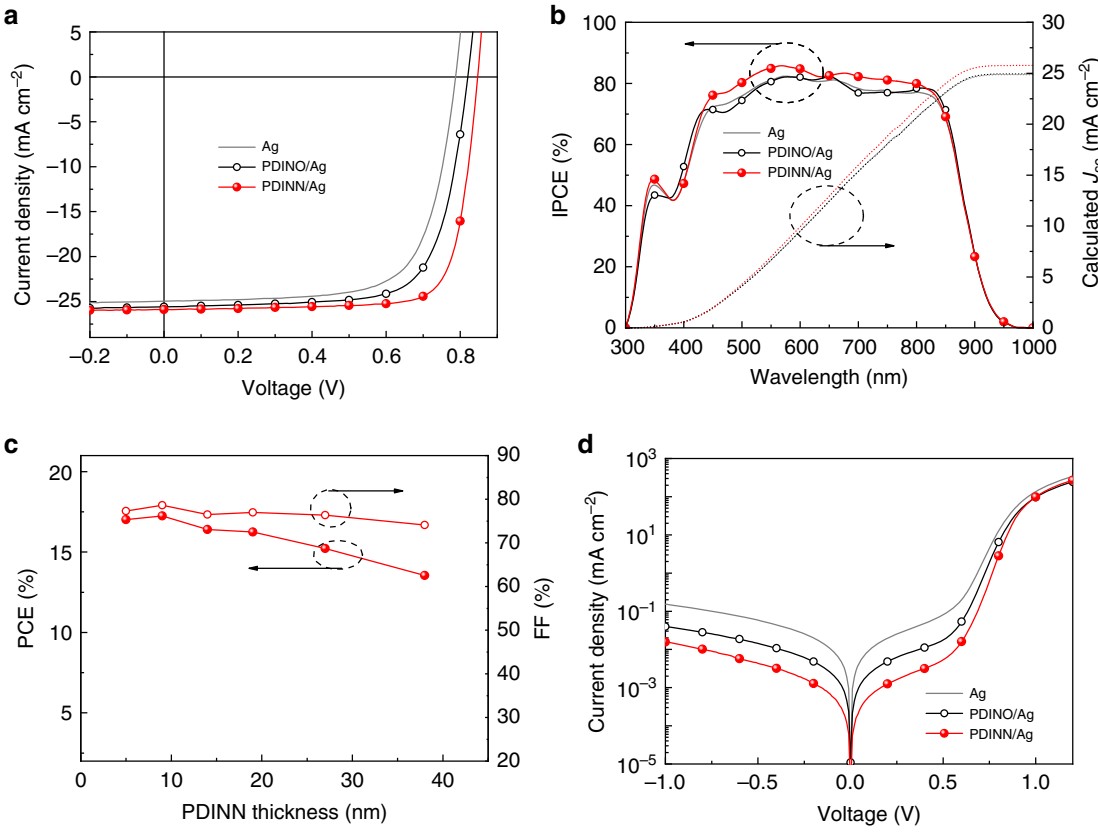

**Fig. 2 Device performance of the OSCs based on PM6:Y6 with Ag as cathode. a** J–V curves of the best OSCs under the illumination of AM 1.5 G, 100 mW cm$^{-2}$. **b** IPCE spectra and integrated current density of the best devices. **c** Dependence of PCE and FF of the OSCs on the thickness of PDINN CIM. **d** Dark currents of the OSCs.

and PFN CIMs that only works well at critical thin thickness (<5 nm)[17,26]. For example, when the PDINN thickness rises to 27 nm, the PCE of the devices still maintains at a high value over 15.0% (Supplementary Fig. 12, Supplementary Table 3). Impressively, the FF still maintains a high value of 74.09% when the PDINN thickness is further increased to 38 nm, indicating the excellent electron-transport and collection ability of the PDINN CIM. Supplementary Fig. 12c, d shows the effect of CIM thickness on the photovoltaic performance of the PDINO CIM-based OSCs. The PCE decreased from 15.46% for the optimized device with PDINO thickness of 9 nm to 11.72% for the OSC with PDINO thickness of 39 nm. Notably, the PCE of the OSC with 38 nm thickness PDINN CIM is even comparable to the peak value (15.46%) for the PDINO-based devices at its optimum thickness (9 nm).

Based on the PDINN thickness insensitivity of the PDINN-based OSCs, we also fabricated large area devices of 1 cm$^2$ to evaluate the prospect of the PDINN/Ag cathode in real application. The J–V curve and IPCE spectrum of the large area device are plotted in Supplementary Fig. 13. As a result, a PCE value of 15.82% is obtained, along with a $V_{oc}$ of 0.850 V, a $J_{sc}$ of 25.42 mA cm$^{-2}$ (calculated $J_{sc}$ from IPCE spectrum is 24.95 mA cm$^{-2}$, with a mismatch of 1.85%), and an FF of 73.19%. The performance is a prominent one for the 1 cm$^2$ device[54].

**The effect of CIMs on device performance.** For the two CIMs, we further evaluate the charge carrier recombination behavior of the OSCs, and studied the dependence of $J_{sc}$ on the light intensity (P), as well as the dependence of $V_{oc}$ on P (ref. [55]). The dependence of $J_{sc}$ (and $V_{oc}$) on P is shown in Fig. 3a. The relationship

between $J_{sc}$ and P can be expressed as $J_{sc} \propto P^\alpha$ (ref. [56]). The value of $\alpha$ should be 1 if all the free carriers were extracted by the electrode[55]. In our cases, the $\alpha$ value of the PDINO-based device is 0.963, while for the PDINN-based device, its $\alpha$ value is increased to 0.982 that is very close to 1. This means there is effective carrier collection and well-suppressed bimolecular recombination in the OSCs with PDINN as CIM. Figure 3a shows the dependence of $V_{oc}$ on light intensity P. The slope of $V_{oc}$ versus ln(P) line should be $kT/q$ for bimolecular recombination[57], while the competition between bimolecular and Shockley– Read–Hall type, trap-assisted recombination makes the slope between $kT/q$ and $2\ kT/q$. The OSCs with methanol treatment (without CIM) and with PDINO or PDINN CIM exhibit the slopes of 1.42, 1.24, and 1.14 $kT/q$, respectively. It is obvious that the slope of the device with PDINN CIM is closer to $kT/q$, which indicates the suppression of the trap-assisted recombination in the devices. The results collectively suggested that charge recombination was effectively suppressed in the PDINN-based OSCs.

Furthermore, we performed transient absorption (TA) measurements in a timescale up to tens of microseconds to study the effect of the CIM on the dynamics of photo-excited charges decay in the OSCs. The experimental details are described in "Methods" section. For the static transport measurements[10], the average diffusion distance at a delay time of 20 ns can be estimated to be on the order of 10 nm at room temperature with no voltage applied. The diffusion length may be much longer if the time-dependent local charge mobility is considered[58,59]. Thus for our samples, excited carrier can diffuse to the active layer surface, and the related probe can give some information on the carrier behavior that affects device performance. On the timescale of tens

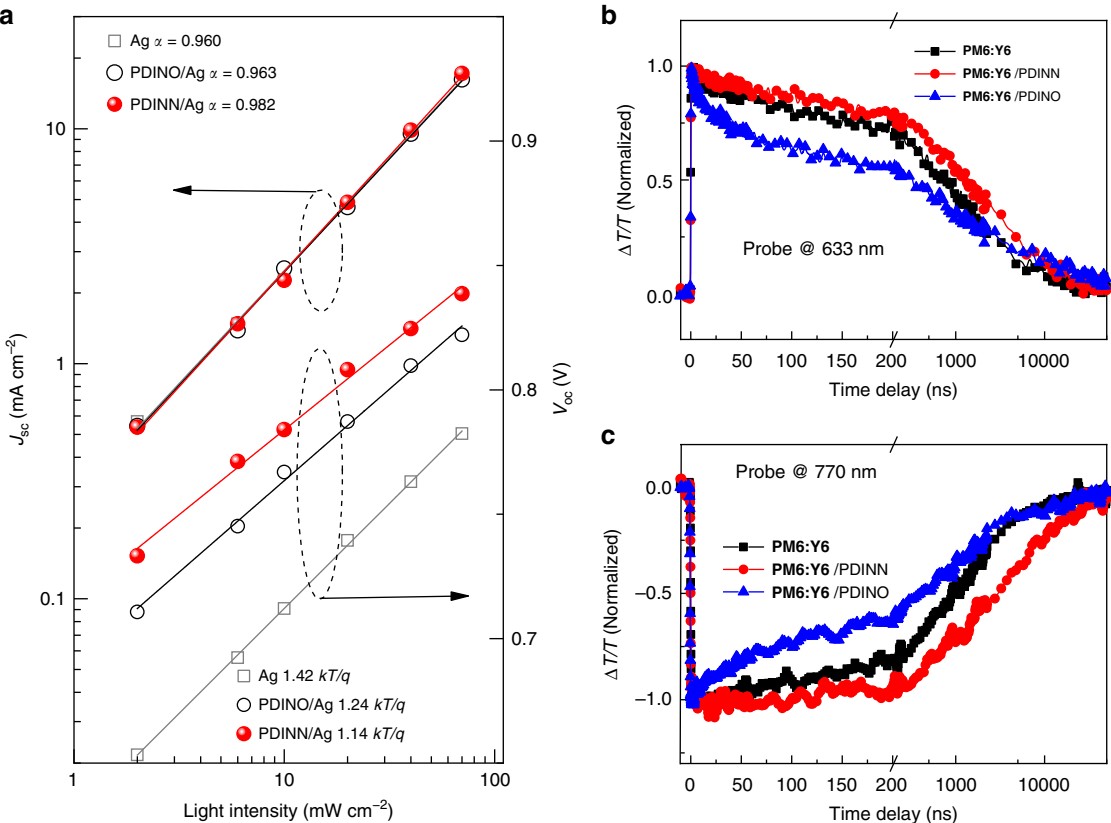

**Fig. 3 The investigation of the charge carrier recombination behavior at the contact. a** Light intensity dependence of $J_{sc}$ and $V_{oc}$ of the OSCs based on PM6:Y6. Free charge dynamics of PM6:Y6/PDINN, PM6:Y6/PDINO, and PM6:Y6 films **b** at 633 nm probe and **c** at 770 nm probe, respectively.

of ns and longer, the TA data are mainly induced by the charge-separated state of free polarons as confirmed by the photo-induced-absorption measurements (Supplementary Fig. 14). In the PM6:Y6 blend, the charge-separated state is configured with electron polaron at the acceptor site and hole polaron at the donor site, which may be captured by the excited-state absorption features of the free polarons, and the ground-state bleach features of donors and acceptors (Supplementary Fig. 14). With increasing pump fluence, the decay of free polarons becomes faster due to bimolecular recombination process (Supplementary Fig. 15). In comparison with the dynamics in the sample with PDINO, the charge recombination in the sample with PDINN is slower at the early stage (Fig. 3b, c), implying that the CIM of PDINN suppresses some loss channel.

For the different dynamics in the two CIM-treated samples, although we have no further evidence to tell how the PDI interlayers affect the charges that diffuse to the surface. The longer-lived photo-excited charges observed in PDINN-treated blend extend the charge lifetime, which is beneficial for charge collection, and thus probably responsible for the improved performance in the PDINN-treated OSCs. Typically, charge recombination is directly related with the key device parameter FF of the devices, the suppressed carrier recombination, and enhanced charge collection in the PDINN-based devices is well consistent, with its higher FF value (78.59%) in comparison with that (72.24%) for the PDINO-based device.

The $J$–$V$ characteristics of the OSCs at dark, (see Fig. 2d) show a slightly large rectification ratio thus a better diode quality for the device with PDINN CIM. To better understand the improved FF of the OSC with PDINN CIM, the electrochemical impedance spectroscopy (EIS) measurements were performed to examine the interface resistance of the devices[60]. Supplementary Fig. 16

shows Nyquist plots of the OSCs with PDINN or PDINO CIM at dark. A bias voltage equal to $V_{oc}$ was applied to dissipate the total current. The data were fitted using the equivalent-circuit model. The series resistances ($R_{series}$) of the PDINN-based device (0.62 Ω cm²) is significantly smaller than that of the PDINO-based device (280.80 Ω cm²). The lower series resistance should be beneficial for the higher FF of the PDINN-based devices.

**Device performance with other active layers**. With the successful application of PDINN CIM in the PM6:Y6-based OSCs, we further studied the universality of the PDINN CIM for the OSCs with other active layers, including PTQ10: IT-4F (ref. [61]) and J11: *m*-ITTC (ref. [62]; Supplementary Figs. 17 and 18). Using Ag as the top cathode, the direct comparison of the PCEs of the OSCs with different CIMs (PDIN/Ag in Supplementary Fig. 19) and different photoactive layers are shown in Fig. 4a, and the related photovoltaic parameters are collected in Supplementary Table 4. Clearly, the PDINN-based devices exhibit better PCEs in all the OSCs, with different active layers compared with that of the corresponding PDINO-based devices, indicating that PDINN is a universal and effective CIM for various OSCs. In consideration that all of the investigated active layers bearing halogen atom or carbonyl group, the formation of the hydrogen bond inter-molecular interactions between the active layer and PDINN is also plausible. Our results also suggest that constructing inter-molecular interactions between the active layer and the electrode interlayer is an effective way for further improving photovoltaic performance, and realizing application of the OSCs.

**Device stability and effect of cathodes on device performance**. At the optimal CIM thickness, we also explored high WF Cu as

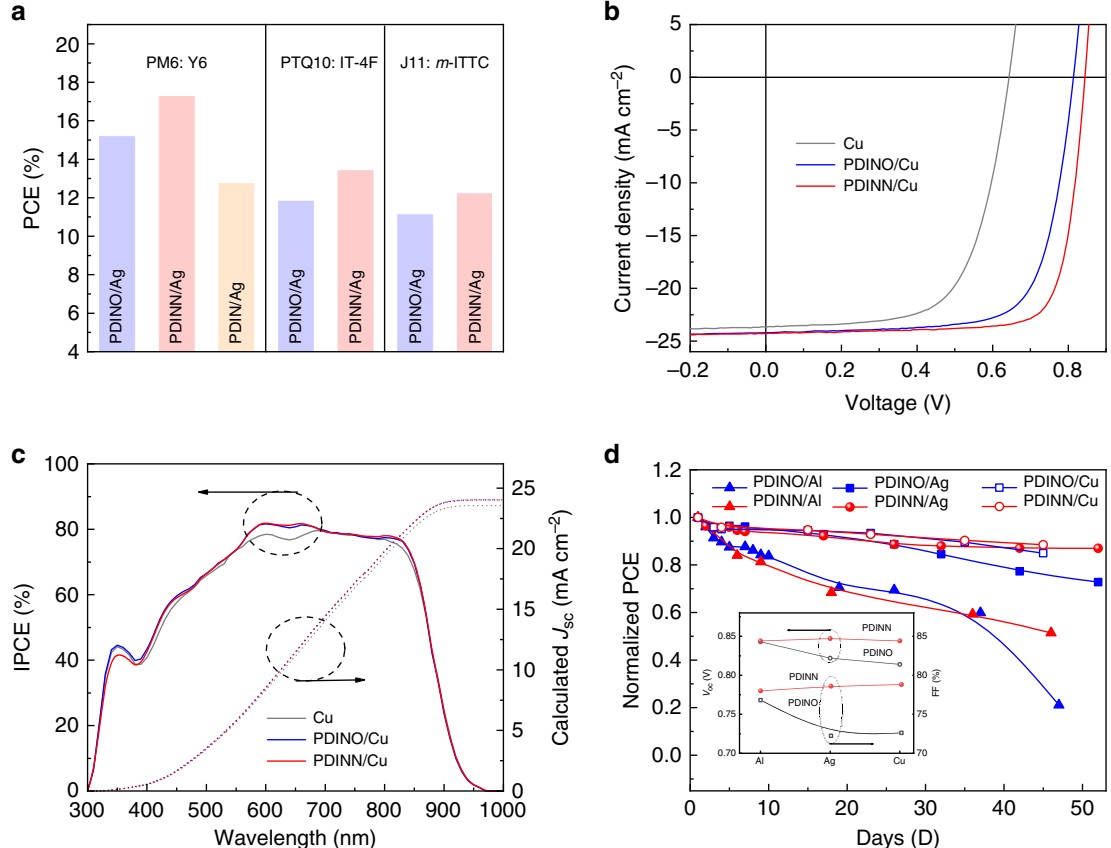

**Fig. 4 Photovoltaic performance of the OSCs base on PM6:Y6. a** The comparison of the PCEs of the OSCs with different CIMs and different photoactive layers under the illumination of AM 1.5 G,100 mW cm$^{-2}$. **b** J–V curves of the best OSCs based on PM6: Y6 blend with copper as cathode under the illumination of AM 1.5 G,100 mW cm$^{-2}$. **c** IPCE spectra and the integrated current density of the best devices with Cu as cathode. **d** Stability of the OSCs with different top cathode of PDINN/Ag or PDINN/Cu. The devices kept in nitrogen environment and in dark between the J–V measurements.

cathode of the OSCs based on PM6: Y6 blend to take its advantage of high conductivity and low price. The J–V curves and IPCE spectra of the OSCs are shown in Fig. 4. Notably, with the PDINN CIM, $V_{oc}$ value (0.844 V) of the Cu-based devices is similar to that of the Ag-based devices (0.847 V). Impressively, a high PCE exceed 16% and a high FF of 78.8% are obtained for the PDINN/Cu-based device, indicating that Cu is also a good choice for cheap and stable top electrode in future commercialization of the OSCs, with PDINN as CIM. The lower IPCE value in the wavelength region from 300 to 600 nm is associated with the low reflectivity of the Cu-based device (Supplementary Fig. 20), which explains the slightly lower $J_{sc}$ value with Cu as cathode. In addition, the result suggested that a higher $J_{sc}$ value thus higher PCE could be obtained by suitable optical management[34] in the Cu electrode-based OSCs, with PDINN as CIM.

The stability of the interlayer itself and interlayer/electrode contact is important issue for real application of OSCs. With the good air and thermal stability of PDINN, we further investigated its device stability using Ag or Cu as top electrode. After ~50 days, the PCEs of the PDINN/Ag-based and PDINN/Cu-based devices can retain 87 and 88% of their initial PCE values in a glove box filled with nitrogen (Fig. 4d), while using PDINO as CIM, the PCEs remained 73 and 85% of their initial PCEs for the Ag-based and Cu-based devices, respectively. Supplementary Fig. 21a displays the steady-state efficiency of the OSCs (in a glove box filled with nitrogen) under continuous air mass (AM) 1.5 G (100 mW cm$^{-2}$) illumination. The primary results over 600 s show a better stability of the PDINN-based devices (98.21% of

initial efficiency) than that of the PDINO-based devices (95.90% of initial efficiency). For the unencapsulated device in air (Supplementary Fig. 21b, c), the PDINN/Ag and PDINN/Cu devices maintained 93.5% and 86.3% of their initial PCE value, whereas the PDINO/Ag and PDINO/Cu devices only showed 35.2% and 70.0% of their initial efficiency after storing 150 h. The big difference between the stability of PDINN and PDINO-based devices is closely related to the CIMs used. The low thermal stability and moisture sensitive of PDINO can lead to an irreversible damage to the CIM–cathode contact, which renders the device stability poor. All the above results indicate that the PDINN-based OSCs show better stability than that of the PDINO-based devices.

To better understand the effect of the top electrodes (Al, Ag, and Cu) and the CIMs on the device performance, corresponding device parameters are summarized in Supplementary Tables 5 and 6 for comparison (J–V curves and IPCE spectra of the Al-based devices are shown in Supplementary Fig. 22). From Supplementary Table 5, it can be seen that for the PDINO-based devices with cathodes from Al, Ag to Cu, the $V_{oc}$ values, and FF values are gradually decreased due to a limited WF tunability for PDINO, they are 0.843 V and 76.81% for Al, 0.821 V and 73.52% for Ag, and 0.814 V and 72.61% for Cu, respectively. In contract, the PDINN-based devices demonstrated almost unchanged $V_{oc}$ (ca. 0.844 V) and FF (ca. 78.0 %) values for all the OSCs with Al, Ag, and Cu cathode (Supplementary Table 6) benefited from a strong WF tunability of PDINN. In addition, the advantage of Ag electrode with strong reflectivity and low resistance should also

account for its higher PCE values over that of the PDINO-based OSCs. The trends of variation in $V_{oc}$ and FF are plotted in the inset of Fig. 4d for a clear comparison. The lower WF of the cathode increases the built-in electric field of the device, which is beneficial to increasing charge extraction efficiency and reducing recombination losses for higher FF of the PDINN-based OSCs. The results suggested that PDINN is an excellent CIM for future application of the OSCs with high WF stable metals as cathode.

## Discussion

An aliphatic amine-functionalized PDI derivative, PDINN, is synthesized and investigated as CIM in OSCs. Compared with widely used CIM of PDINO, the advantages of PDINN are its better contact with non-fullerene active layers, stable electrode interface, higher conductivity, and stronger ability to reduce WFs of the metal cathode, which make it more suitable for use as CIM in non-fullerene OSCs with air stable metal cathode, such as Ag and Cu, to improve the device stability. With PDINN/Ag as the top electrode, the OSCs based on PM6:Y6 exhibits a high PCE of 17.23% (certified PCE of 16.77% by NREL), which is one of the highest efficiency reported for the single-junction binary OSCs. The high efficiency is mainly benefited from its high FF of 78.59%, which can be understood by the good contact with active layer and lower WF of the cathode of the OSCs. Notably, in combination with PDINN, the devices with Ag cathode not only yield higher efficiency, but also exhibit high stability. In addition, PDINN can be easily synthesized by one-step reaction with high yield and large-scale product (over 64 g in lab) from cheap raw materials. Thus, PDINN is a low-cost and high-performance CIM for OSCs, and is a promising CIM for future large-scale roll-to-roll production and commercial application of the OSCs. Also, PDINN could also be used as cathode interlayer in other thin-film optoelectronic devices. In addition, our result also indicates that modifying the interfacial contact with suitable intermolecular interactions is a feasible approach to improve the device performance in thin-film electronic devices.

## Methods

**Materials and synthesis**. Y6 was purchased from eFlexPV, and PM6 was purchased from Solarmer Materials.

Synthesis of PDINN: A mixture of perylene-3,4,9,10-tetracarboxylic dianhydride (39.2 g, 100.0 mmol) and N,N-dimethyldipropylenetriamine (64.1 g, 400.0 mmol) in methanol (300 mL) was stirred at reflux for 8 h to form a clear red solution. After removal of the solvent under reduced pressure, the residue was purified by column chromatography, followed by precipitation from acetone to give PDINN as red solid (64.4 g, 95.5%). The NMR data are in agreement with the reference[63]. $^1$H NMR (400 MHz, CDCl$_3$), δ (ppm): 8.40 (s, 4H,), 8.23 (t, 4H), 4.24 (t, 4H), 2.75 (t, 4H), 2.69 (t, 4H), 2.33 (t, 4H), 2.22 (s, 12H), 1.96(t, 4H), 1.68 (m, 6H). $^{13}$C NMR (75 MHz, CDCl$_3$) δ 162.86, 133.62, 130.69, 128.66, 125.45, 122.86, 122.55, 58.01, 48.30, 47.31, 45.56, 38.65, 28.40, 28.18. HRMS (MALDI-TOF): calcd for C$_{40}$H$_{46}$N$_6$O$_4$, 674.3581, found, 674.3630.

**Material characterization**. $^1$H NMR and $^{13}$C NMR spectra were recorded on Bruker AVANCE 400 MHz or Bruker AVANCE 300 MHz NMR spectrometer at room temperature. Mass spectra were measured on a Shimadzu spectrometer. TGA was measured on a Perkin-Elmer TGA-7 thermogravimetric analyzer with a heating rate of 10 °C min$^{-1}$ under a nitrogen flow rate of 100 mL min$^{-1}$. The UV–vis absorption spectra were measured by Hitachi U-3010 UV–vis spectrophotometer. The thin-film samples were prepared by spin-coating their methanol solutions on quartz plates. To measure the thickness of thin films, several grooves were marked on the thin film by pointed tweezer, and then the depths of the grooves were recorded on Bruker DEKTAK XT step profiler. The average value of the depths was used as the thickness of thin film. Cyclic voltammetry was conducted on a Zahner IM6e electrochemical workstation using sample film coated on glassy carbon as the working electrode, Pt wire as the counter electrode, and Ag/AgCl as the reference electrode, in a 0.1 M tetrabutylammonium hexafluorophosphate (Bu$_4$NPF$_6$) acetonitrile solution and ferrocene/ferrocenium (Fc/Fc$^+$) couple was used as an internal reference. EIS measurements were performed on Zahner IM6e electrochemical workstation, and a bias voltage equal to $V_{oc}$ was applied to dissipate the total current. The UPS spectra were recorded on ESCALab250Xi multifunction X-ray photoelectron spectrometer. The ESR spectra were recorded at

298 K using a JES-FA2000 ESR Spectrometer. The GIWAXS measurements were conducted at PLS-II 6 A U-SAXS beamline of the Pohang Accelerator Laboratory in Korea. The ATR FT-IR spectra was recorded on BRUKER-Fourier Transform Infrared Spectrometer-TENSOR 27. The information about contact angle and Owen method for calculating the surface energy could be found in Supplementary Method.

**Morphology measurement by PiFM**. The morphology of PDINN and PDINO on the active layer of PM6:Y6 was examined using PiFM (refs. [64,65]). The PiFM measurements were performed on a VistaScope AFM. Firstly, the infrared (IR) spectra of the PDINN and PDINO CIMs, and the photovoltaic materials PM6 and Y6 were measured, as shown in Supplementary Fig. 5a. Then the characteristic IR peak at 1653 cm$^{-1}$ for PDINN and PDINO was selected for the PiFM morphology measurement of PDINN and PDINO films on the PM6:Y6 active layer, as shown in Supplementary Fig. 5b. Because PM6 and Y6 don't have the IR peak at 1653 cm$^{-1}$, the PiFM images show only the morphology of the CIMs without the influence of PM6:Y6 active layer below it. It can be seen from Supplementary Fig. 5b, PDINO molecules aggregate and form an uneven film, while PDINN film is more uniform due to the hydrogen bonding between PDINN and the active layer.

**Conductivity measurements**. Solutions of PDINO and PDINN in chloroform were spin-coated at a film thickness ($T$) of ~100 nm on substrates with parallel silver electrodes (thickness = 100 nm). The length of the electrode ($W$) was 11,000 μm and the distance ($L$) between the two electrodes was 500 μm. Electrical characterization was conducted using a PDA FS380. The conductivity was extracted from the equation of $\sigma = I \times L/(V \times W \times T)$.

**Device fabrication and characterization of the OSCs**. The OSCs were fabricated with a structure of ITO/PEDOT:PSS/active layer/CIM/Al, Ag or Cu. The ITO glass was cleaned by sequential ultrasonic treatment in water, deionized water, acetone and isopropanol, and then treated in an ultraviolet ozone cleaner (Ultraviolet Ozone Cleaner, Jelight Company, USA) for 20 min. The PEDOT:PSS aqueous solution (Baytron P 4083 from H. C. Starck) was filtered through a 0.45 mm filter and then spin-coated on precleaned ITO-coated glass at 4000 rpm for 30 s. Subsequently, the PEDOT:PSS film was annealed at 150 °C for 20 min in air to form a 30 nm film. A blend solution of PM6 donor and Y6 acceptor was prepared by dissolving the materials in chloroform, and then was spin-coated at 4000 rpm onto the PEDOT:PSS layer. Then methanol solution of CIMs (PDINO and PDINN) at a concentration of 1.0 mg mL$^{-1}$ was deposited on the active layer at 3000 rpm for 30 s to afford a cathode buffer layer. Finally, the metal cathode Al, Ag, or Cu was thermal evaporated under a mask at a base pressure of ~10$^{-5}$ Pa. The photovoltaic area of the device is 4.6 mm$^2$. Optical microscope (Olympus BX51) was used to define the active area of the devices. The $J$–$V$ characteristics of the OSCs were measured in a nitrogen glove box with a Keithley 2450 Source Measure unit. Oriel Sol3A Class AAA Solar Simulator (model, Newport 94023A) with a 450 W xenon lamp and an AM 1.5 filter was used as the light source. The light intensity was calibrated to 100 mW cm$^{-2}$ by a Newport Oriel 91,150 V reference cell. The voltage step and delay time were 10 mV and 1 ms, respectively. The scan started from −1.5 V to 1.5 V. The IPCE was measured by Solar Cell Spectral Response Measurement System QE-R3-011 (Enli Technology Co., Ltd., Taiwan). The light intensity at each wavelength was calibrated with a standard single-crystal Si photovoltaic cell.

**TA experiment**. For ns-TA experiment, the pump beam used in the experiment was emitted from a pico-second laser diode (LDH-P-C-670M, PicoQuant GmbH) at 633 nm or 770 nm. The pump fluence was ~1 μJ cm$^{-2}$ and the carrier density could be estimated as ~2 × 10$^{17}$ cm$^{-3}$, which is close to the lowest pump condition as mentioned in the reference[66]. The active layers of the samples are spin-coated directly on glass substrates without top electrodes. The probe light was a supercontinuum white light generated by focusing a portion of the femtosecond laser beam from a regeneration amplifier (Libra, Coherent) onto a 3-mm-thick sapphire plate. The synchronization and time difference between the pump and probe was controlled by a digital delay generator (DG645, Stanford Instruments). The overall temporal resolution was better than 0.5 ns. The TA signal was analyzed by an InGaAs photo-diode array (G11608, Hamamatsu) with shot to shot detection at 5 kHz enabled by a home-built FPGA control board. The samples were kept at nitrogen atmosphere during the measurements to avoid photodegradation.

**Reporting summary**. Further information on research design is available in the Nature Research Reporting Summary linked to this article.

## Data availability

The data that support the findings of this study are available from the corresponding author on request.

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

## Acknowledgements
The work was supported by the National Natural Science Foundation of China (nos. 51722308, 51673200, 21734008, and 51820105003), Guangdong Major Project of Basic and Applied Basic Research (No. 2019B030302007), Fundamental Research Funds for the Central Universities (Buctrc201822 and XK1802-2), and the Key Research Programs in Universities of Henan Province (18A150049). The authors thank Prof. Wang Xiuyu and Dr. Zhang Jinyuan for their valuable discussions.

## Author contributions
Y.F.L. and Z.G.Z. supervised the project. Z.G.Z. designed PDINN, Z.G.Z. and L.W.X. synthesized PDINN and PDINO, L.W.X. characterized the chemical structure of PDINN and PDINO, J.Y. and B.B.Q. carried out the device fabrication and characterization, L.M. analyzed the device parameters, S.S.C. and C.Y. measured the GIWAXS diffraction patterns, Q.J.Z. confirmed the H-bonding material of PDINN, C.K.S. synthesized PTQ10, R.W., C.F.Z., and M.X. measured TA spectra, and provide discussions. Z.G.Z., J.Y., and Y.F.L. wrote the paper.

## Competing interests
The authors declare no competing interests.
