## [Peer Review File · Nature Communications]

Reviewers' comments:

Reviewer #1 (Remarks to the Author):

In this manuscript, the authors developed a new cathode interlayer material (PDINN) consisting of aliphatic amine side chains and a perylene diimide core, which can down-shift the work function of metal cathodes including Al, Ag, and Cu. Organic solar cells (OSCs) with PDINN as interlayer and Ag as cathode demonstrated a high power conversion efficiency (PCE) of 17.23% and high stability based on the hot PM6:Y6 system. The utilization of polar group functionalized conjugated materials, which was often nominated as water/alcohol soluble conjugated materials for improving the performance of organic optoelectronic devices including OSCs is not new. In the past ten years, numerous materials that featured polar groups (amine, phosphate, sulfate, and PEG etc.) and conjugated backbone (either electron-rich or electron-deficient) for use as electrode modifiers in solar cells have been reported in literatures. In this manuscript, the authors demonstrated considerable device performance enhancement based on their newly developed interface material PDINN, which is even superior to their previously reported cathode modification material PDINO. The prominent device results prove that PDINN is indeed a good modifier for a few high work function metals. Nevertheless, several critical scientific issues were not correctly explained in the current manuscript.

1) An important starting point and critical claim of this paper is that their interface material can provide good adhesion via chemical interaction with the active layer (emphasized in the INTRODUCTION part). However, there is no solid evidence to support this argument, which makes the discussion too speculative. The only supportive results are the variable-temperature ¹H-NMR of the blended solutions consisting of PDINN and Y6. This is not convincing since one cannot use the results obtained in solutions to support the statement for solid states. Moreover, one cannot take it for granted that the improved device performance was offered by the improved adhesion since there are many other possibilities. I would suggest the authors conducting ¹H-NMR and FTIR/Raman measurements using the samples in solid state to convincingly evidence the chemical interactions such as Hydrogen-bonding, which is indispensable for the new statement described in the manuscript.

2) The authors ascribed the enhanced FF of the solar cells to the improved adhesion between interface layer and active layer, which lacks convincing evidences as I pointed out above. On the other hand, the function of enhanced electrical conductivity was underestimated, which would actually be the main reason for the improved device performance in my opinion. It is well known that high conductivity and the ability to form Ohmic contact is critical to an efficient interface material, which can promote charge extraction and improve FF. In this manuscript, much stronger electron spin resonance signal was observed for PDINN with respect to PDINO (Figure 1f). This is a clear implication for the more efficient self-doping and higher conductivity of PDINN than PDINO (please see the papers of Prof. Alex Jen and Prof. Yong Cao). Therefore, more experiments are needed to correlate the relationship between high device performance and high conductivity of interlayer materials. Meanwhile, the relevant discussion may need revisions.

3) When studying the stability of devices based on PDINN interlayer material, the authors studied the air and thermal stability. It is also necessary to study the light stability of the devices, especially under ultraviolet light. The information about the device stability upon continuous illumination would be very helpful to prove the good adhesion of the interface layer.

4) The authors did not show device stability data of Al cathode device in Figure 4(d). Is it because the stability of Al cathode device is poor? And from Figure 4(d), we can find that the stability of the PDINO/Cu device is similar to that of PDINN/Cu device. Therefore, one cannot state that PDINN can improve the device stability compared to PDINO.

5) When PDINO or PDINN is used as the interlayer materials, the work function of Al is lower than that of Ag and Cu. And the performance of devices with Al as cathode is better than Ag and Cu for PM6:Y6/PDINO contact. Why did the Ag cathode afford the best performance instead of Al for the PM6:Y6/PDINN devices?

6) Please explain that why PDINN displays a blue-shifted absorption relative to that of PDINO (Figure 1c). Does this correlate to the device performance?

Generally, I would be happy to see the publication of this manuscript in Nature Communication if the above-mentioned issues could be reasonably addressed.

Reviewer #2 (Remarks to the Author):

The manuscript by Yao et al. introduces a new cathode interlayer, aliphatic amine functionalized perylene diimide (PDINN). The PDINN interlayer complements the limits in cathode interlayer based on PDI, such as PDINO. The PDINN is much lower cost due to one step approach from cheap raw materials and has good solubility without the assistance of any acids. In OPVs devices, the PDINN has better adhesion with organic active layers due to hydrogen bonding from aliphatic amine group. This better adhesion enhances an ability in lowering work function of cathode, leading to higher built-in potential and higher VOC. This better ability in lowering work function makes it possible to replace Al cathode with Ag cathode which is more stable in the air. Secondary, the better adhesion also improves charge extraction and reduce recombination, leading to higher FF. As a result, using a PDINN cathode interlayer, the PCE of PM6/Y6 device exceeded 17% with better performance, stability and low cost than the traditional cathode interlayer, PDINO.

The authors have applied the new cathode interlayer to improve quality of devices. Recently, there was a similar report which introduced new anode interlayer and reached to 17% of PCE [1]. Therefore, this report, introducing the cathode interlayer, does not seem to have noticeable novelty in terms of the efficiency of devices. However, with the better performance, the PDINN have an amazing attraction because of the low cost, simple step for synthesis even and good compatibility with stable cathode. These characteristics may provide a strong tool to many researchers and the PDINN might speed up development in solar cells researches.

[1] Lin et al., Adv. Mater. 2019, 31, 1902965

A few questions and comments are as follows:

2. Authors said "However, this approach is constrained by the intrinsic high surface tension of the interlayer bearing the polar group, which harms its deposition with a solution processed layer-by-layer technology and results in a poor interlayer contact with the active layer". The authors are advised to provide the surface tension of the PDINN and PDINO.

3. In figure 2c, the authors mention thickness insensitivity of the PDINN. However, there is no data to prove superiority of PDINN thickness insensitivity. PCE vs. PDINO thickness of OSC devices, would be needed.

4. The (1 0 0) peak representing face on stacking is important for improving OSC devices. However, the (1 0 0) peak intensity of PDINN is lower than PDINO. Comments on the results would be needed.

5. "With the good air and thermal stability of PDINN, we further investigated its device stability using Ag or Cu top electrode. After about 50 days, the PCEs of the PDINN / Ag-based and PDINN / Cu-based devices can retain 87% and 88% of their initial PCE value in a glove box filled with nitrogen (Figure 4d)". If the authors want to show the device air stability, the experimental data in air environment should be provided as well.

Reviewer #3 (Remarks to the Author):

Cathode interlayers are important to develop efficient and stable organic solar cells. For low-cost interlayers, they are generally insulate materials that are not suitable for practical application. The incorporation of aromatic compounds can enhance their conductivity, but this approach is associated with tedious purification steps, thus the interlayer can only accessible with high cost. This is an obstacle encountered for large-scale production and industrial application. In this manuscript, the authors reported a new low-cost interlayer (PDINN) based on perylene-diimide that can be facile synthesized by a typical one-step approach under a large-scale of 64 g in lab from cheap raw materials. PDINN offers stable electrode interface, high conductivity, effective work function tuning with air stable metals (Ag and Cu), and good contact with the nonfullerene active layer under a hydrogen-bond interaction, thus making the device working well with a record high efficiency of 17.23% with Ag (certified value of 16.77% by NREAL). Thus, PDINN is a low-cost (1.6 \$ g⁻¹) and effective interfacial engineering material that is prominent for thin film photovoltaic devices and cable of large-scale production. Most specially, the approach of using noncovalent-bond interaction to improve of the adhesion of the interlayer on the active layer is interesting and a new concept to design new interlayer materials. This work can attract broad interest attract broad interest for both academia and industry. The followings are some suggested minors issues.

1. To decrease the work function of cathode, generally, a highly polar groups is introduced to increase its dipole moment. As the authors point out, this approach is usually constrained by the poor adhesion of the interlayer on the active layer brought by the polar group. That is why fewer reported interlayer worked efficiently with air stable metals that can be widely used in different active layers. In Page 18, the authors tried different active layers to confirm that PDINN is a universal interlayers that can be widely used. When make discussions, can the authors correlated this advantage with this special amine groups?
2. Thermal stability is an important issue for real application, and the authors have given some evidence that PDINN shows better stability. Also, thermogravimetric analysis should be carried out to further evaluate the thermal stabilities.
3. In Figure S5, from morphology, the authors give a strong evidence that PDINN does show a better adhesion on the active layers than that of PDINO. This observation together with the variable-temperature and titration ¹H NMR experiments (Figure S6) shows that intermolecular interaction can help PDINN layer form a good contact with the actively layers during its layer-by-layer deposition. To help the readers had a better understand on the conclusion, more details explanation on the experimental results of PDINN (Fig 5Sd) and PDINO (Fig 5Sb) should be given.

Reviewer #4 (Remarks to the Author):

The authors present a novel PDI derivative bearing aliphatic amine sidechains (denoted as PDINN) as a cathode interlayer material for highly efficient organic solar cells. Compared to the commonly employed PDIDO, the new material stands out by its ability to form hydrogen-bonds with the active organic layer, thereby improving the adhesion between the layers. Also, stable metals such as Au or Ag coated with PDINN exhibit work functions of ca. 4 eV, which renders these modified metals suited for electron-injecting/extracting contacts. Notably, self-doping results in a fairly high conductivity of the PDINN layers, making the device performance less sensitive to the thickness of the PDINN. As a consequence, highly efficient OSCs are fabricated from different polymer:NFA blends with stable Ag

and Cu electrodes.

The performance data presented in the work are indeed impressive as is the tolerance of the newly developed cathode interlayer material versus the choice of the cathode metal and the processing conditions. The material is clearly superior to PDINO, which is now widely applied for the production of state of the art NFA blends (e.g. J. Yuan, Y. Zhang, L. Zhou, G. Zhang, H.-L. Yip, T.-K. Lau, X. Lu, C. Zhu, H. Peng, P.A. Johnson, M. Leclerc, Y. Cao, J. Ulanski, Y. Li, and Y. Zou, *Joule* 3, 1140 (2019)). I, therefore, believe that the results presented in this work will have a large impact on the future research the OPV community. I also appreciate that the authors do a great job to justify most of their claims by providing detailed data on the polymer and device characteristics (e.g. the ability of hydrogen bonding to the photoactive material by performing variable temperature and titration NMR spectroscopy). Therefore, I recommend publication of this work.

Nevertheless, revision is mandatory, in particular regarding the analysis of the physical and device properties.

The authors analyze the dark JV-characteristics of their devices to deduce values of the series and shunt resistances and the ideality factor. This approach is based on the modified Shockley-diode equation and it is very common for the study of inorganic semiconductors, which because of doping and high carrier mobilities exhibit high electrical conductivities. This presumption is not fulfilled in organic semiconducting layers as has been conclusively demonstrated by several groups in the past (see e.g. Figure 3 in K. Tvingstedt and C. Deibel, *Adv. Energy Mater.* 6, 1502230 (2016) or the work by U. Würfel, D. Neher, A. Spies, and S. Albrecht, *Nat. Commun.* 6, 6951 (2015)). Therefore, this analysis and the corresponding interpretation (e.g. the calculated V_{oc}) needs to be removed from the paper.

The information provided on the TAS measurements is insufficient and it is, therefore, totally unclear which conclusion can be drawn from these data: what was the pump fluence:

- how large is the carrier density at this excitation condition?
- why was the probe wavelength 633 nm, which excitation is dominating the signal at this wavelength and why did you use an IR-sensitive InGaAs to measure at this wavelength?
- what was the architecture of the measured sample (was the layer on PEDOT or on glass, was there a top electrode or not)?

I am also concerned with the claim that “the PDINN CIM heals surface traps”. Which surface traps? On the PM6:Y6 surface? Or at the interface to the CIM?

Clearly, results from additional measurements on a neat PM6:Y6 need to be provided, measurements have to be performed at different fluence in order to identify the order of recombination (see e.g. 1 I.A. Howard, R. Mauer, M. Meister, and F. Laquai, *J. Am. Chem. Soc.* 132, 14866 (2010)) and the transient spectra have to be displayed.

I am concerned with several of the claims on page 16 (and the corresponding graphs in Figure 3).

- Why should an p-n junction form between the active PM6:Y6 blend and the CIM. The CIM is (probably n-doped) but the active layer is not p-doped (at least, I did not find any evidence for p-doping in the manuscript)?
- What crucial information can be drawn from the PiFM image in Figure (e)? How relevant is this image for the understanding of the function of the interlayer?
- The authors conclude from the results in Figure (f) that “that the interlayer can also serve as the light harvest layer when the interlayer is in contact with the donor domain due to the charge transfer effect from donor to interlayer”. First, the photocurrents in Figure (f) are very small but also, the authors do not provide spectral information which would be needed to draw conclusions

about the origin of the photocurrent

- The author claim that the rectification behavior in Figure (g) is induced “by an interfacial dipole layer due to the interaction between the amine group of interlayers with electron-deficient part of the n-OS acceptors”. But given the fact that for these devices, the Y6 acceptor layer is sandwiched between a high workfunction PEDOT:PSS anode and a low workfunction CIM/Ag cathode, rectification is expected anyhow. This is because PEDOT:PSS does not inject electrons and CIM/Ag does not inject holes. In my opinion, the shown rectification behavior has nothing to do with an interfacial dipole layer but it is simply related to the difference in work function between the top and bottom contact.

Other more minor issues:

The abstract is poorly written, with grammar mistakes or missing words (e.g. “requires a highly polar groups” or “offering effective down-shifting work function of air stable cathodes”)

The authors demonstrate in Supplementary Figure 4 that PDINN is more stable under prolonged heating (compared to PDINO) but they don't explain why this is the case.

In Supplementary Figure 5, the authors employ PiFM to show the superior wetting ability of PDINN on PM6:Y6. However, as the technique is not properly introduced (PiFM is not a common technique in the OPV field), it is not clear what is actually shown in the graphs. For example, the image in Figure (b) differs from that in Figure (c) by the larger feature size and the larger signal range. Which information is used to conclude that film formation of PDINO on PM6:Y6 is poor?

The authors employ lateral JV experiments to determine the conductivity of the interlayer. However, the JVs from these measurements in Supplementary Figure 7 are plotted on a log-lin scale. It's therefore, not obvious that the current is indeed linear in voltage as would be expected for purely ohmic behavior.

Figure 2 shows the JVs of the best devices while table 2 provides the average values with standard deviations. Nevertheless, a graph with the full device statistics must be provided as well (e.g. as a Supplementary Figure)

On page 15, the text reads: “The OSCs with methanol treated (without CIM) and with PDINN or PDINO CIM exhibit the slopes of 1.42, 1.24 and 1.19 kT/q, respectively” which is not consistent with the numbers of Figure 3(a).

More information needs to be provided on the device stability measurement. Have the samples been kept in the dark or under illumination between the JV measurements? Unfortunately, neither the caption nor the text provides these details. Also, what exactly is shown in the inset of Figure 4(d)?

Response to Reviewer #1:

..... *In this manuscript, the authors demonstrated considerable device performance. Enhancement based on their newly developed interface material PDINN, which is even superior to their previously reported cathode modification material PDINO. The prominent device results prove that PDINN is indeed a good modifier for a few high work function metals. Nevertheless, several critical scientific issues were not correctly explained in the current manuscript.*

1) *An important starting point and critical claim of this paper is that their interface material can provide good adhesion via chemical interaction with the active layer (emphasized in the INTRODUCTION part). However, there is no solid evidence to support this argument, which makes the discussion too speculative. The only supportive results are the variable-temperature ¹H-NMR of the blended solutions consisting of PDINN and Y6. This is not convincing since one cannot use the results obtained in solutions to support the statement for solid states. Moreover, one cannot take it for granted that the improved device performance was offered by the improved adhesion since there are many other possibilities. I would suggest the authors conducting ¹H-NMR and FTIR/Raman measurements using the samples in solid state to convincingly evidence the chemical interactions such as Hydrogen-bonding, which is indispensable for the new statement described in the manuscript.*

Response: Thanks for the reviewer's comments and suggestions! To provide more evidence that supporting the formation of hydrogen-bonding between Y6 and PDINN, we performed ATR FT-IR measurement in solid blend sample of Y6 and PDINN. The results and related discussions were added in pp. 7~8. **“To provide more evidence on the interactions between Y6 in the active layer and PDINN CIM in the OSC devices, we performed Attenuated Total Reflectance (ATR) FT-IR measurement for the solid blend sample of Y6 and PDINN (w/w,**

1:1), and the result is shown in Supplementary Figure 7a. In contrast to the ATR FT-IR spectrum of pure Y6 sample with the C–F mean peak at 1350 cm^{-1} , the C–F peak of the Y6/PDINN blend sample moves to 1297 cm^{-1} and is broadened, which indicates the formation of the hydrogen-bond between secondary amine group of PDINN and C-F group of Y6 in the solid state.”

Moreover, we designed new experiment to confirm the interaction between PDINN and the solid Y6 molecules by measuring the ^{19}F NMR spectra of Y6. The result and related discussions are added in p. 8: “Moreover, we further studied the interaction between PDINN and solid Y6 by measuring ^{19}F NMR spectra of solid Y6 power in D-methanol NMR tubes with and without PDINN. The ^{19}F NMR spectra of the samples are shown in Supplementary Figure 7b. It can be seen that there is a ^{19}F signal at -152 ppm in the PDINN D-methanol solution, which is from the F atom of Y6. However, for the control sample without PDINN in D-methanol, there is no such signal for Y6 because Y6 is not soluble in methanol. The result suggests that PDINN can provide intermolecular interaction with the organic semiconductor (especially those bearing fluorine groups⁴) such as Y6 during the deposit of PDINN from its methanol solution.”

In addition, we agree with the reviewer that the good adhesion is not the only reason for the good device performance. The factor of high conductivity is also important, and this factor was discussed in the next response.

2) *The authors ascribed the enhanced FF of the solar cells to the improved adhesion between interface layer and active layer, which lacks convincing evidences as I pointed out above. On the other hand, the function of enhanced electrical conductivity was underestimated, which would actually be the main reason for the improved device performance in my opinion. It is well known that high conductivity and the ability to form Ohmic contact is critical to an efficient interface material, which can promote charge extraction and improve FF. In this manuscript, much stronger electron spin resonance signal was observed for PDINN with respect to PDINO (Figure 1f). This is a clear implication for the more efficient self-doping and higher conductivity of PDINN than PDINO (please see the papers of Prof. Alex Jen and Prof. Yong Cao). Therefore, more experiments are needed to correlate the relationship between high device performance and high conductivity of interlayer materials. Meanwhile, the relevant discussion may need revisions.*

Response: We agree with the reviewer that the high conductivity is an important factor for the good device performance for PDINN CIM. Actually, we measured the conductivities of the CIMs and the conductivity of PDINN is much higher than that of PDINO, as mentioned in the bottom of p. 10: “The conductivities of PDINO and PDINN calculated from the I - V curves (Supplementary Figure 9) are 2.4×10^{-6} S/cm and 5.0×10^{-4} S/cm, respectively, which indicate that PDINN has a better electron transport ability.” To further clarify this factor, we measured the ac impedance of the OSCs, and the results and related discussion were added in p. 17~18: “To better understand the improved FF of the OSC with PDINN CIM, the electrochemical impedance spectroscopy (EIS) measurements were performed to examine the interface resistance of the devices. Supplementary Figure 16 shows Nyquist plots of the OSCs with PDINN or PDINO CIM at dark. A bias voltage equal to V_{oc} was applied to dissipate the total current. The data were fitted using the equivalent-circuit model. The series resistances (R_{series}) of the PDINN-based device ($0.62 \Omega \text{ cm}^2$) is significantly smaller than that of the PDINO-based device ($280.80 \Omega \text{ cm}^2$). The lower series resistance should be beneficial for the higher FF of the PDINN-based devices.”

3) *When studying the stability of devices based on PDINN interlayer material, the authors studied the air and thermal stability. It is also necessary to study the light stability of the devices, especially under ultraviolet light. The information about the device stability upon continuous illumination would be very helpful to prove the good adhesion of the interface layer.*

Response: According to the reviewer’s suggestion, we designed experiment to study the device stability operated at the maximum power voltage under continuous AM 1.5G illumination. Related result was added in p.20. “Supplementary Figure 21a displays the steady-state efficiency of the OSCs (in a glovebox filled with nitrogen) under continuous AM1.5G (100 mWcm^{-2}) illumination. The primary results over 600 s show a better stability of the PDINN-based devices (98.21% of initial efficiency) than that of the PDINO-based devices (95.90% of initial efficiency).” In consideration that the AM 1.5G spectrum also have the ultraviolet part, so the result obtained above can also help to understand the stability under ultraviolet light.

4) *The authors did not show device stability data of Al cathode device in Figure 4(d). Is it because the stability of Al cathode device is poor? And from Figure 4(d), we can find that the stability of the PDINO/Cu device is similar to that of PDINN/Cu device. Therefore, one*

cannot state that PDINN can improve the device stability compared to PDINO.

Response: The device stability data of Al cathode device were added in Figure 4d. Under our tested conditions, it is true that the trend of the normalized stability of the PDINO/Cu device is similar to that of PDINN/Cu device under nitrogen atmosphere. However, for the unencapsulated device in ambient air, as shown in Supplementary Figure 21 b and c, the PDINN-based devices are more stable than PDINO-based devices. Based on the result, related discussion was added in p. 20. “For the unencapsulated device in air (Supplementary Figure 21b and 21c), the PDINN/Ag and PDINN/Cu devices maintained 93.5% and 86.3% of their initial PCE value, whereas the PDINO/Ag and PDINO/Cu devices only showed 35.2% and 70.0% of their initial efficiency after storing 150 h.”

5) *When PDINO or PDINN is used as the interlayer materials, the work function of Al is lower than that of Ag and Cu. And the performance of devices with Al as cathode is better than Ag and Cu for PM6:Y6/PDINO contact. Why did the Ag cathode afford the best performance instead of Al for the PM6:Y6/PDINN devices?*

Response: It is generally accepted that the top Ag cathode with the advantages of strong reflectivity and high conductivity can help to improve the photo current of the photovoltaic cells. This conclusion can be supported by the higher J_{SC} of the Ag cathode devices (25.89 mA cm⁻² for PDINN/Ag, 25.28 mA cm⁻² for PDINO/Ag) in comparison with that of the Al cathode devices (24.41 mA cm⁻² for PDINN/Al, 24.55 mA cm⁻² for PDINO/Al) and the Cu cathode devices (24.28 mA cm⁻² for PDINN/Cu, 24.22 mA cm⁻² for PDINO/Cu) (see Supplementary Table 5 and Supplementary Table 6).

6) *Please explain that why PDINN displays a blue-shifted absorption relative to that of PDINO (Figure 1c). Does this correlate to the device performance?*

Response: We think the blue-shifted absorption of PDINN than that of PDINO could be related to the hydrogen bonding property of PDINN. We measured the hydrogen bonding property of PDINN by FT-IR measurement and added several sentences in p. 9 to explain the phenomenon: “To explain the phenomenon of the blue-shifted absorption, we measured the hydrogen bonding property of PDINN in comparison with PDIN³³ by measure the ATR FT-IR spectra of PDINN and PDIN, as shown in Supplementary Figure 7c. The carbonyl vibration at 1692 cm⁻¹ for PDIN is shifted to 1687 cm⁻¹ for PDINN, indicating the hydrogen bonding formation in PDINN. Therefore, the blue-shifted absorption of PDINN film could be

related to the different aggregation of PDINN molecules in comparison with the PDINO molecules due to its intramolecular hydrogen bonding property.⁴³” In addition, as we known, the intra- and or inter-molecular interaction affect the packing of the PDI core as revealed by XRD patterns, which is a possible factor related with the device performance.

Response to Reviewer #2:

..... *The authors have applied the new cathode interlayer to improve quality of devices. Recently, there was a similar report which introduced new anode interlayer and reached to 17% of PCE [1]. Therefore, this report, introducing the cathode interlayer, does not seems to have noticeable novelty in terms of the efficiency of devices. However, with the better performance, the PDINN have an amazing attraction because of the low cost, simple step for synthesis even and good compatibility with stable cathode. These characteristics may provide a strong tool to many researchers and the PDINN might speed up development in solar cells researches.*

1) *Authors said “However, this approach is constrained by the intrinsic high surface tension of the interlayer bearing the polar group, which harms its deposition with a solution processed layer-by-layer technology and results in a poor interlayer contact with the active layer”. The authors are advised to provide the surface tension of the PDINN and PDINO.*

Response: Our purpose to write the sentences is to emphasize the poor adhesion of the CIM with strong polar substituents like PDINO on the nonfullene-based active layer of OSCs, and to need the design of new CIMs with strong polar substituents and good adhesion on the active layer. Here we designed PDINN with hydrogen bonding ability with the photovoltaic materials in the active layer as discussed in pp. 7~8. And PDINN possesses good adhesion with the active layer.

2) *In Figure 2c, the authors mention thickness insensitivity of the PDINN. However, there is no data to prove superiority of PDINN thickness insensitivity. PCE vs. PDINO thickness of OSC devices, would be needed.*

Response: We have evaluated the PDINO thickness sensitivity according to the Reviewer’s comments, the results are shown in Supplementary Figure 12, and the related discussion was added in p. 14. “**Supplementary Figure 12c,d show the effect of CIM thickness on the**

photovoltaic performance of the PDINO CIM-based OSCs. The PCE decreased from 15.46% for the optimized device with PDINO thickness of 9 nm to 11.72% for the OSC with PDINO thickness of 39 nm. Notably, the PCE of the OSC with 38 nm thickness PDINN CIM is even comparable to the peak value (15.46%) for the PDINO based devices at its optimum thickness (9 nm).”

3) *The (1 0 0) peak representing face on stacking is important for improving OSC devices. However, the (1 0 0) peak intensity of PDINN is lower than PDINO. Comments on the results would be needed.*

Response: Following the suggestion, the *d*-spacing and coherence length estimated from the in-plane (010) and out-of-plane (100) diffraction of PDINO and PDINN are provided in Supplementary Figure 8 c and d. And related comments were added in p. 10. “With a smaller terminal group in PDINO, although it shows a stronger aggregation behavior, its predominantly edge-on orientation as revealed by the strong (100) peak in the out-of-plane direction may weaken its carrier extraction ability from the active layer.”

4) *“With the good air and thermal stability of PDINN, we further investigated its device stability using Ag or Cu top electrode. After about 50 days, the PCEs of the PDINN/Ag-based and PDINN/Cu-based devices can retain 87% and 88% of their initial PCE value in a glove box filled with nitrogen (Figure 4d)”. If the authors want to show the device air stability, the experimental data in air environment should be provided as well.*

Response: We also evaluated the device air stability according to the Reviewer’s suggestion. Based on the result, related discussion was added in p. 20. “For the unencapsulated device in air (Supplementary Figure 21b and 21c), the PDINN/Ag and PDINN/Cu devices maintained 93.5% and 86.3% of their initial PCE value, whereas the PDINO/Ag and PDINO/Cu devices only showed 35.2% and 70.0% of their initial efficiency after storing near 150 h.”

Response to Reviewer #3:

..... Thus, PDINN is a low-cost (1.6 \$ g⁻¹) and effective interfacial engineering material that is prominent for thin film photovoltaic devices and cable of large-scale production. Most specially, the approach of using noncovalent-bond interaction to improve of the adhesion of the interlayer on the active layer is interesting and a new concept to design new interlayer materials. This work can attract broad interest attract broad interest for both academia and industry. The followings are some suggested minors issues

1) *To decrease the work function of cathode, generally, a highly polar groups is introduced to increase its dipole moment. As the authors point out, this approach is usually constrained by the poor adhesion of the interlayer on the active layer brought by the polar group. That is why fewer reported interlayer worked efficiently with air stable metals that can be widely used in different active layers. In Page 18, the authors tried different active layers to confirm that PDINN is a universal interlayer that can be widely used. When make discussions, can the authors correlate this advantage with this special amine groups?*

Response: Thanks for the reviewer's comments and revision suggestions! Following the comments and suggestion, related discussion was added in p. 18. “**In consideration that all of the investigated active layers bearing halogen atom or carbonyl group, the formation of the hydrogen bond intermolecular interactions between the active layer and PDINN is also plausible. Our results also suggest that constructing intermolecular interactions between the active layer and the electrode interlayer is an effective way for further improving photovoltaic performance and realizing application of the OSCs.**”

2) *Thermal stability is an important issue for real application, and the authors have given some evidence that PDINN shows better stability. Also, thermogravimetric analysis should be carried out to further evaluate the thermal stabilities.*

Response: We performed the thermogravimetric analysis of PDINN and PDINO under an inert atmosphere to further evaluate their thermal stability, and the results are shown in Supplementary Figure 4b. The related discussion was added in p. 7: “**The good thermal stability of PDINN is further confirmed by their thermogravimetric analysis, which show the 5% weight-loss temperature at 103.4 °C for PDINIO and 240.1 °C for PDINN (Supplementary Figure 4b).**”

3) *In Figure S5, from morphology, the authors give a strong evidence that PDINN does show a better adhesion on the active layers than that of PDINO. This observation together with the variable-temperature and titration ¹H NMR experiments (Figure S6) shows that intermolecular interaction can help PDINN layer form a good contact with the actively layers during its layer-by-layer deposition. To help the readers had a better understand on the conclusion, more details explanation on the experimental results of PDINN (Fig 5Sd) and PDINO (Fig 5Sb) should be given.*

Response: Following the reviewer’s comments and suggestions, the related discussion was

put in the section of "**Morphology measurement by PiFM**" in Supplementary Information.

“The morphology of PDINN and PDINO on the active layer of PM6:Y6 was examined using photoinduced force microscopy (PiFM).^{1,2} Firstly the infrared (IR) spectra of the PDINN and PDINO CIMs and the photovoltaic materials PM6 and Y6 were measured, as shown in Supplementary Figure 5(a). Then the characteristic IR peak at 1653 cm^{-1} for PDINN and PDINO was selected for the PiFM morphology measurement of PDINN and PDINO films on the PM6:Y6 active layer, as shown in Supplementary Figure 5(b). Because PM6 and Y6 don't have the IR peak at 1653 cm^{-1} , the PiFM images show only the morphology of the CIMs without the influence of PM6:Y6 active layer below it. It can be seen from Supplementary Figure 5(b), PDINO molecules aggregate and form an uneven film, while PDINN film is more uniform due to the hydrogen bonding between PDINN and the active layer.”

Response to Reviewer #4:

The authors present a novel PDI derivative bearing aliphatic amine sidechains (denoted as PDINN) as a cathode interlayer material for highly efficient organic solar cells. Compared to the commonly employed PDIDO, the new material stands out by its ability to form hydrogen-bonds with the active organic layer, thereby improving the adhesion between the layers. Also, stable metals such as Au or Ag coated with PDINN exhibit work functions of ca. 4 eV, which renders these modified metals suited for electron-injecting/extracting contacts. Notably, self-doping results in a fairly high conductivity of the PDINN layers, making the device performance less sensitive to the thickness of the PDINN. As a consequence, highly efficient OSCs are fabricated from different polymer:NFA blends with stable Ag and Cu electrodes. The performance data presented in the work are indeed impressive as is the tolerance of the newly developed cathode interlayer material versus the choice of the cathode metal and the processing conditions. The material is clearly superior to PDINO, which is now widely applied for the production of state of the art NFA blends (e.g. J. Yuan, Y. Zhang, L. Zhou, G. Zhang, H.-L. Yip, T.-K. Lau, X. Lu, C. Zhu, H. Peng, P.A. Johnson, M. Leclerc, Y. Cao, J. Ulanski, Y. Li, and Y. Zou, Joule 3, 1140 (2019)). I, therefore, believe that the results presented in this work will have a large impact on the future research the OPV community. I also appreciate that the authors do a great job to justify most of their claims by providing detailed data on the polymer and device characteristics (e.g. the ability of hydrogen bonding

to the photoactive material by performing variable temperature and titration NMR spectroscopy). Therefore, I recommend publication of this work. Nevertheless, revision is mandatory, in particular regarding the analysis of the physical and device properties.

1) The authors analyze the dark *JV*-characteristics of their devices to deduce values of the series and shunt resistances and the ideality factor. This approach is based on the modified Shockley-diode equation and it is very common for the study of inorganic semiconductors, which because of doping and high carrier mobilities exhibit high electrical conductivities. This presumption is not fulfilled in organic semiconducting layers as has been conclusively demonstrated by several groups in the past (see e.g. Figure 3 in K. Tvingstedt and C. Deibel, *Adv. Energy Mater.* 6, 1502230 (2016) or the work by U. Würfel, D. Neher, A. Spies, and S. Albrecht, *Nat. Commun.* 6, 6951 (2015)). Therefore, this analysis and the corresponding interpretation (e.g. the calculated V_{oc}) needs to be removed from the paper.

Response: Thanks for the reviewer's positive comments and the valuable revision suggestions. Following the reviewer's suggestions, the related discussion on the dark *JV*-characteristics of the devices was removed in the revised manuscript. Instead, we performed the widely used electrochemical impedance spectroscopy measurements to discussion carrier recombination inside the devices. The result and the related discussion were added in p.17~18:

“To better understand the improved FF of the OSC with PDINN CIM, the electrochemical impedance spectroscopy (EIS) measurements were performed to examine the interface resistance of the devices. Supplementary Figure 16 shows Nyquist plots of the OSCs with PDINN or PDINO CIM at dark. A bias voltage equal to V_{oc} was applied to dissipate the total current. The data were fitted using the equivalent-circuit model. The series resistances (R_{series}) of the PDINN-based device ($0.62 \Omega \text{ cm}^2$) is significantly smaller than that of the PDINO-based device ($280.80 \Omega \text{ cm}^2$). The lower series resistance should be beneficial for the higher FF of the PDINN-based devices.”

2) The information provided on the TAS measurements is insufficient and it is, therefore, totally unclear which conclusion can be drawn from these data.

2.1) What was the pump fluence: how large is the carrier density at this excitation condition?

Response: The pump fluence and the carrier density wer provided in the section of "Ns-

transient absorption experiment" in Supplementary Information.

“For ns-transient absorption experiment, the pump beam used in the experiment was emitted from a pico-second laser diode (LDH-P-C-670M, PicoQuant GmbH) at 633 nm or 770 nm. The pump fluence was about $1 \mu\text{J}/\text{cm}^2$ and the carrier density could be estimated as about $2 \times 10^{17}/\text{cm}^3$, which is close to the lowest pump condition as mentioned in the reference. (*JACS* 132, 14866 (2010))”

2.2) *Why was the probe wavelength 633 nm, which excitation is dominating the signal at this wavelength and why did you use an IR-sensitive InGaAs to measure at this wavelength?*

Response: The reason that probing at 633 nm was given in p. 15: “We adopt a broadband probe (600 nm-1100 nm) for the samples with the TA spectra shown in Supplementary Figure 14. It can be seen that the maximal TA signal of the charge separated state appears at 633 nm, so we chose this probe wavelength to maximize the signal to noise ratio.”

The reason of using an IR-sensitive InGaAs to measure was given in the section of "**Ns-transient absorption experiment"** in Supplementary Information. “To maintain good sensitivity in the probe range of $> 950 \text{ nm}$, we used an InGaAs photodiode array (G11608, Hamamatsu) which had wide spectral response range from 500 nm to 1700 nm. In this case, we can adopt a broadband probe (600 nm-1100 nm) for the PM6:Y6, PM6:Y6/PDINN and PM6:Y6/PDINO films at different time delays in the TAS measurements.”

2.3) *What was the architecture of the measured sample (was the layer on PEDOT or on glass, was there a top electrode or not)? I am also concerned with the claim that “the PDINN CIM heals surface traps”. Which surface traps? On the PM6:Y6 surface? Or at the interface to the CIM?*

Response: The architecture of the measured sample was provided in the section of "**Ns-transient absorption experiment"** in Supplementary Information: “The active layers of the samples are spin-coated directly on glass substrates without top electrodes.” As for the surface traps, the related sentence was revised to " the healing of the active layer surface traps with PDINN CIM." (p. 16)

2.4) *Clearly, results from additional measurements on a neat PM6:Y6 need to be provided, measurements have to be performed at different fluence in order to identify the order of recombination (see e.g. I I.A. Howard, R. Mauer, M. Meister, and F. Laquai, J. Am. Chem.*

Soc. 132, 14866 (2010)) and the transient spectra have to be displayed.

Response: Following the reviewer's suggestion, we performed additional TA measurements on a neat PM6:Y6 film, and did the experiments at different fluence. The related results are discussed in pp. 15~16: "The recorded charge dynamics of the samples are shown in Figure 3b. The charge separated state in neat PM6:Y6 film shows slightly shorter lifetime than PM6:Y6/PDINN film and decays much slower than that in PM6:Y6/PDINO film, which indicates the traps at the active layer interface to the CIM dominates the loss of charge carriers in PM6:Y6/PDINO film. At different pump fluence, the charge decay dynamics of PM6:Y6/PDINN and PM6:Y6/PDINO films are compared in Supplementary Figure 15 at 633 nm probe. The charge lifetime of the samples increases with lower pump fluence, which is consistent with the bimolecular recombination process of charge separated states. In all the pump conditions, the carrier life time of PM6:Y6/PDINO film is much shorter than that of PM6:Y6/PDINN, from which we can safely rule out the fluence induced faster free charge decay in PM6:Y6/PDINO film."

3) *I am concerned with several of the claims on page 16 (and the corresponding graphs in Figure 3.) Why should a n-p-n junction form between the active PM6:Y6 blend and the CIM. The CIM is (probably n-doped) but the active layer is not p-doped (at least, I did not find any evidence for p-doping in the manuscript)? What crucial information can be drawn from the PiFM image in Figure (e)? How relevant is this image for the understanding of the function of the interlayer? The authors conclude from the results in Figure (f) that "that the interlayer can also serve as the light harvest layer when the interlayer is in contact with the donor domain due to the charge transfer effect from donor to interlayers interlayer". First, the photocurrents in Figure (f) are very small but also, the authors do not provide spectral information which would be needed to draw conclusions about the origin of the photocurrent. The author claim that the rectification behavior in Figure (g) is induced "by an interfacial dipole layer due to the interaction between the amine group of interlayers with electron-deficient part of the n-OS acceptors". But given the fact that for these devices, the Y6 acceptor layer is sandwiched between a high workfunction PEDOT:PSS anode and a low workfunction CIM/Ag cathode, rectification is expected anyhow. This is because PEDOT:PSS does not inject electrons and CIM/Ag does not inject holes. In my opinion, the shown rectification behavior has nothing to do with an interfacial dipole layer but it is simply related to the difference in work function between the top and bottom contact.*

Response: In the original manuscript, the model of p-n junction that used to study the interfacial contact was adopted base on the method described in two literatures (*Joule* 3, 227-239 (2019); *Energy Environ. Sci.*10, 1784-1791 (2017)). In our manuscript, firstly, we used PiFM to check whether the donor rich region or acceptor rich region can be formed in the active layer surface. We think that in this case the model mentioned in the above references can be well established. In our PiFM images, we did find donor rich region and acceptor rich region. And according to the model proposed, in the donor rich region, a diode of PEDOT:PSS/PM6: CIM/Ag was formed, so it is true for an acceptor rich region. At this stage, to address the concerns of the reviewers raised on the p-n junction, more experiments should be conducted, and this is out of the scope of this manuscript.

Therefore, based on the fact that the functions of the interlayers were well investigated and conclusion is soundly, this part was removed in the revised manuscript.

4) *The abstract is poorly written, with grammar mistakes or missing words (e.g. “requires a highly polar groups” or “offering effective down-shifting work function of air stable cathodes”)*

Response: We have carefully checked the abstract part, corrected the grammar mistakes and smoothed the sentences.

5) *The authors demonstrate in Supplementary Figure 4 that PDINN is more stable under prolonged heating (compared to PDINO) but they don't explain why this is the case.*

Response: We performed the thermogravimetric analysis of PDINN and PDINO to further evaluate their thermal stability with the results shown in the supporting information. And related discussion was added in p. 7. “**The good thermal stability of PDINN is further confirmed by their thermogravimetric analysis, which show the 5% weight-loss temperature at 103.4 °C for PDINO and 240.1 °C for PDINN (Supplementary Figure 4b).**”

6) *In Supplementary Figure 5, the authors employ PiFM to show the superior wetting ability of PDINN on PM6:Y6. However, as the technique is not properly introduced (PiFM is not a common technique in the OPV field), it is not clear what is actually shown in the graphs. For example, the image in Figure (b) differs from that in Figure (c) by the larger feature size and the larger signal range. Which information is used to conclude that film formation of PDINO on PM6:Y6 is poor?*

Response: The measurement method and the result of the PiFM measurement were described in the section of "**Morphology measurement by PiFM**" in Supplementary Information: "The morphology of PDINN and PDINO on the active layer of PM6:Y6 was examined using photo induced force microscopy (PiFM).^{1,2} Firstly, the infrared (IR) spectra of the PDINN and PDINO CIMs and the photovoltaic materials PM6 and Y6 were measured, as shown in Supplementary Figure 5(a). Then the characteristic IR peak at 1653 cm^{-1} for PDINN and PDINO was selected for the PiFM morphology measurement of PDINN and PDINO films on the PM6:Y6 active layer, as shown in Supplementary Figure 5(b). Because PM6 and Y6 don't have the IR peak at 1653 cm^{-1} , the PiFM images show only the morphology of the CIMs without the influence of PM6:Y6 active layer below it. It can be seen from Supplementary Figure 5(b), PDINO molecules aggregate and form an uneven film, while PDINN film is more uniform due to the hydrogen bonding between PDINN and the active layer."

7) *The authors employ lateral JV experiments to determine the conductivity of the interlayer. However, the JVs from these measurements in Supplementary Figure 7 are plotted on a log-lin scale. It's therefore, not obvious that the current is indeed linear in voltage as would be expected for purely ohmic behavior.*

Response: Following the Reviewer's suggestion, we have re-plotted the data, the result shows that the current is linear in voltage for ohmic behavior in Supplementary Figure 9b,c where the currents show linear relationship with the voltages. The reason we plotted the data on a log-lin scale is to show the conductivity difference between PDINN and PDINO in a clear way.

8) *Figure 2 shows the JVs of the best devices while table 2 provides the average values with standard deviations. Nevertheless, a graph with the full device statistics must be provided as well (e.g. as a Supplementary Figure).*

Response: According to the reviewer's suggestion, we have provided the histogram of the PCE counts based the full device statistics in Supplementary Figure 10, as indicated in a sentence in p. 13: "The efficiency histograms of PDINN-based and PDINO-based devices are provided in Supplementary Figure 10."

9) *On page 15, the text reads: "The OSCs with methanol threated (without CIM) and with PDINN or PDINO CIM exhibit the slopes of 1.42, 1.24 and 1.19 kT/q, respectively" which is not consistent with the numbers of Figure 3(a).*

Response: We have checked manuscript and corrected this mistake, see p. 15: " The OSCs with methanol threated (without CIM) and with PDINO or PDINN CIM exhibit the slopes of 1.42, 1.24 and 1.19 kT/q, respectively."

10) *More information needs to be provided on the device stability measurement. Have the samples been kept in the dark or under illumination between the JV measurements? Unfortunate, neither the caption nor the text provides these details. Also, what exactly is shown in the inset of Figure 4(d)?*

Response: Yes, our samples have been kept in nitrogen environment and in dark between the $J-V$ measurements. For clarify, the measurement condition was added in the caption of Figure 4d: "The devices kept in nitrogen environment and in dark between the $J-V$ measurements". As for the inset of Figure 4(d), the original description is wrong. We have revised the description in p. 21. "From Supplementary Table 5, it can be seen that for the PDINO-based devices with cathodes from Al, Ag to Cu, the V_{oc} values and FF values are gradually decreased, they are 0.843 V and 76.81% for Al, 0.821 V and 73.52% for Ag, and 0.814 V and 72.61% for Cu respectively. In contract, the PDINN-based devices demonstrated almost unchanged V_{oc} (ca. 0.844 V) and FF (ca. 78.0 %) values for all the OSCs with Al, Ag and Cu cathode (Supplementary Table 6), so that showing higher PCE values over that of the PDINO-based OSCs. The stronger ability of PDINN CIM in lowering the WFs of cathode relative to PDINO should account for the higher V_{oc} and higher J_{sc} values of the PDINN-based OSCs. The trends of variation in V_{oc} and FF are plotted in the inset of Figure 4d for a clear comparison."

Reviewers' comments:

Reviewer #1 (Remarks to the Author):

After revision, several critical scientific issues in my previous review have been explained in the revised manuscript, including the demonstration of the chemical interaction between Y6 and PDINN, reasons for the enhanced FF acquired by PDINN devices, and the device stability issues. There are only a few minor modifications remaining to be done before the paper being formally published, which are as follows.

(1) The authors mentioned that 'This phenomenon is likely related to the hydrogen bond formed between the secondary amine in the aliphatic amine groups of PDINN with the active layer surface as suggested by the variable-temperature and titration ^1H NMR experiments conducted with PDINN and Y6 in solution (Supplementary Figure 6). The ^1H NMR signal at $\delta 1.54$ (assigned to hydrogen in the secondary amine) broadened and downfield shifted, which provide a clear evidence that PDINN forms a stronger hydrogen-bond with Y6 in the solution' (page 7, line 140-145). In fact, the ^1H -NMR signal at $\delta 1.54$ in Supplementary Figure 6a may be assigned to the trace H_2O in solvent CDCl_3 . While adding small portion of PDINN in the Y6/ CDCl_3 solution, signal at $\delta 1.54$ broadened and shifted to low field, this could result from the Hydrogen bonding either between PDINN and trace H_2O or between PDINN and Y6. So, it is better to combine the following (ATR) FT-IR measurements and ^{19}F NMR spectra characterizations to draw the conclusion instead of directly drawing conclusion like that in (page 7, line 140-145).

(2) The (ATR) FT-IR measurements and ^{19}F NMR spectra characterizations in the revised manuscript (page 7-8, line 146-160) is positive to prove the possible chemical Hydrogen bonding between PDINN and Y6. I would thus suggest the authors directly comparing the FT-IR spectra of PDINN and PDINN:Y6 in the Supplementary Information to more clearly demonstrate the difference and to prove the possible chemical Hydrogen bonding PDINN and Y6.

(3) The authors added that the air stability of PDINN-based devices were much more superior to their PDINO counterparts (page 20, line 386-389), please explain a little bit about this phenomenon. Also, it is better to give the detailed test environment of the air stability in this paper, such as the temperature and relative humidity.

(4) Among the top electrodes mentioned in this manuscript, Al electrode-based devices show the best performance when adopting PDINO as CIM (Supplementary Table 4), while Ag electrode-based device stand out when using PDINN CIM (Supplementary Table 5). These intriguing results still lack reasonable explanations. This issue has been pointed out as Q5 in my previous review comments, but was not reasonably explained in the revised manuscript.

Reviewer #2 (Remarks to the Author):

Regarding the response 2-1),

Contrary to the PDINO, the PDINN has enhanced adhesion even with the strong polar substituents and this factor seems to be the most important in your report. Therefore, you need to show exact quantitative or numerical data on the adhesion degree of PDINO and PDINN. The adhesion is correlated to surface tension as you mentioned. You can get 'work of adhesion' using 'Owens-Wendt model' [1].

[1] L.Q.N.Tran et al., Understanding the interfacial compatibility and adhesion of natural coir fibre thermoplastic composites, *Compos Sci Technol* 80, 23-30 (2013).

This reviewer was satisfied with the other responses. Thank you.

Reviewer #3 (Remarks to the Author):

I read the response letter and feel that the authors have addressed the comments from previous reviewers.

Reviewer #4 (Remarks to the Author):

The authors did a great job in taking into account most of the issues raised by the reviewers. Therefore, the manuscript may be suited for publication in Nature Communications provided that a remaining issue can be properly addressed.

I am still not convinced by the description of the TA experiments and the conclusions drawn from these experiments for the following two reasons.

Assignment of the transient absorption spectra: According to the newly added Figure S14, the transient absorption spectra consist of two spectral regions where the pump increases the transparency of the sample (at 600 nm – 670 nm and between 820 nm and 880 nm), while there is a pump-induced decrease of the transmission in the remaining part of the spectrum. It is often proposed that an increased transmission is due to a ground state bleach (GSB), which is expected to show its maximum signal where the ground state absorption is largest. This may explain the strong positive signal at ca. 637 nm (where PM6 shows a strong absorption) but not of the second positive peak at 850 nm (because Y6 has its absorption maximum at 800 nm). On the other hand, a photoinduced decrease of transmission is by the absorption of newly created photogenerated species, such as excitons, polarons etc (excited state absorption, ESA). The text assigns the peak at 633 nm to a charge separated state and that at 770 nm to “the ESA” signal. What exactly is meant with charge separated state and why does it cause the peak at 637 nm. Why did the authors chose to plot the ESA at 780 nm, and why is the temporal decay of the ESA (apparently) different at e.g. 950 nm? The authors are asked to extend the discussion of the spectral features in the TA spectra and their specific decay properties.

Interpretation of the transient properties: The transient data in Figure 3b,c and in Figure S15 display a continuous decrease of the signal over several microseconds, and this decrease becomes accelerated at higher fluence. I agree with the authors that this long term decay is dominated by bimolecular recombination. But what is not clear to me is the sudden (early) decay of the transient signal in the sample with PDINO. The authors propose that this decay is due to trapping of charges at surface traps. I see two problems with this interpretation. First, if the signal at 633 nm stems from the ground state bleach on PM6, why would trapping of charges (by e.g. impurities and surface defects) restore the original PM6 absorption properties. Instead, trapping would slow down the return of the photogenerated electron back to the polymer backbone, thereby prolonging the signal decay. Second, the timescale of this early decay is 10-30 ns. If this decay would be due to charge carrier trapping at surface states, a considerable fraction of photogenerated electrons and/or holes must be able to travel to the surface. Since there is no electric field applied, this motion must be due to diffusion. I strongly doubt that the diffusion of electron and holes in the PM:Y6 blend is fast enough to allow them travelling to the surface within few tens of nanoseconds. Therefore, the authors need to carefully reconsider their interpretation.

Response to Reviewer #1:

After revision, several critical scientific issues in my previous review have been explained in the revised manuscript, including the demonstration of the chemical interaction between Y6 and PDINN, reasons for the enhanced FF acquired by PDINN devices, and the device stability issues. There are only a few minor modifications remaining to be done before the paper being formally published, which are as follows.

(1) The authors mentioned that 'This phenomenon is likely related to the hydrogen bond formed between the secondary amine in the aliphatic amine groups of PDINN with the active layer surface as suggested by the variable-temperature and titration ^1H NMR experiments conducted with PDINN and Y6 in solution (Supplementary Figure 6). The ^1H NMR signal at $\delta 1.54$ (assigned to hydrogen in the secondary amine) broadened and downfield shifted, which provide a clear evidence that PDINN forms a stronger hydrogen-bond with Y6 in the solution' (page 7, line 140-145). In fact, the ^1H -NMR signal at $\delta 1.54$ in Supplementary Figure 6a may be assigned to the trace H_2O in solvent CDCl_3 . While adding small portion of PDINN in the $\text{Y6}/\text{CDCl}_3$ solution, signal at $\delta 1.54$ broadened and shifted to low field, this could result from the Hydrogen bonding either between PDINN and trace H_2O or between PDINN and Y6. So, it is better to combine the following (ATR) FT-IR measurements and ^{19}F NMR spectra characterizations to draw the conclusion instead of directly drawing conclusion like that in (page 7, line 140-145).

Response:

Thanks for the reviewer's insightful comments. According to the suggestion, we revised our manuscript as the followings in pp 8-9. "The ^1H NMR signal at $\delta 1.54$ (assigned to hydrogen in the secondary amine) broadened and downfield shifted. To provide more evidence on the interactions between Y6 in the active layer and PDINN CIM in the OSC devices, we performed Attenuated Total Reflectance (ATR) FT-IR measurement for the solid blend sample of Y6 and PDINN (w/w, 1:1), and the result is shown in Supplementary Figure 7a. In contrast to the ATR FT-IR spectrum of pure Y6 sample with the C-F mean peak at 1350 cm^{-1} , the C-F peak of the Y6/PDINN blend sample moves to 1297 cm^{-1} and is broadened. Moreover, we further studied the interaction between PDINN and solid Y6 by measuring ^{19}F NMR spectra of solid Y6 powder in methanol- D_4 NMR tubes with and without PDINN. The ^{19}F NMR spectra of the samples are shown in Supplementary Figure 7b. It can be seen that there is a ^{19}F signal at -152 ppm in the PDINN methanol- D_4 solution, which is from the F atom of Y6. However, for the control sample without PDINN in methanol- D_4 , there is no such signal for Y6 because Y6 is not soluble in methanol. Thus the above results collectively provide strong evidence for the formation of the hydrogen-bond between secondary amine group of PDINN and C-F group of Y6 in the solid state."

(2) *The (ATR) FT-IR measurements and ^{19}F NMR spectra characterizations in the revised manuscript (page 7-8, line 146-160) is positive to prove the possible chemical Hydrogen bonding between PDINN and Y6. I would thus suggest the authors directly comparing the FT-IR spectra of PDINN and PDINN:Y6 in the Supplementary Information to more clearly demonstrate the difference and to prove the possible chemical Hydrogen bonding PDINN and Y6.*

Response: Now the IR spectra of PDINN was also provided in Supplementary Figure 7a for a clear comparison to obtain the conclusion that the formation of hydrogen-bond for PDINN with Y6.

(3) *The authors added that the air stability of PDINN-based devices were much more superior to their PDINO counterparts (page 20, line 386-389), please explain a little bit*

about this phenomenon. Also, it is better to give the detailed test environment of the air stability in this paper, such as the temperature and relative humidity.

Response: The temperature (25 °C) and the relative humidity (40%) was provide in the caption of Supplementary Figure 21. Relate discussion was added in p 20. “**The big difference between the stability of PDIN and PDINO-based devices is closely related to the CIMs used. The low thermal stability and moisture sensitive of PDINO can lead to an irreversible damage to the CIM-cathode contact, which renders the device stability poor.**”

(4) Among the top electrodes mentioned in this manuscript, Al electrode-based devices show the best performance when adopting PDINO as CIM (Supplementary Table 4), while Ag electrode-based device stand out when using PDINN CIM (Supplementary Table 5). These intriguing results still lack reasonable explanations. This issue has been pointed out as Q5 in my previous review comments, but was not reasonably explained in the revised manuscript.

Response: When PDINO was used as the interlayer (Supplementary Table 5), because of its limited ability in tune the WF of the top electrodes, the as-fabricated devices can only work well with Al electrode. Attempts to improve the device stability by replacing Al with Ag or Cu for top cathode, commonly result in a lower V_{OC} , FF and J_{SC} of the OSCs, due to a low built-in potential across the device. That is why only PDINO/Al cathode performed best.

While for the PDINN interlayer, it has a more strong ability in tune the WF as shown in Figure 1f and Table 1. This can promise a suitable built-in potential across the device without causing any loss in V_{oc} and FF when Ag was used as electrode. In addition, the special advantage (strong reflectivity and high conductivity) of Ag electrode, can improve the J_{SC} and FF of the photovoltaic cells. That is why Ag electrode performed best when PDINN was used as top electrode. (Supplementary Table 6). So related discussion was revised in pp 20-21.

“From Supplementary Table 5, it can be seen that for the PDINO-based devices with cathodes from Al, Ag to Cu, the V_{oc} values and FF values are gradually decreased due to a limited WF tunability for PDINO, they are 0.843 V and 76.81% for Al, 0.821 V and 73.52% for Ag, and 0.814 V and 72.61% for Cu respectively. In contract, the PDINN-based devices demonstrated almost unchanged V_{oc} (ca. 0.844 V) and FF (ca. 78.0 %) values for all the OSCs

with Al, Ag and Cu cathode (Supplementary Table 6) benefited from a strong WF tunability of PDINN. In addition, its advantage of Ag electrode with strong reflectivity and low resistance should also account for its higher *PCE* values over that of the PDINO-based OSCs.”

Response to Reviewer #2:

Regarding the response 2-1), Contrary to the PDINO, the PDINN has enhanced adhesion even with the strong polar substituents and this factor seems to be the most important in your report. Therefore, you need to show exact quantitative or numerical data on the adhesion degree of PDINO and PDINN. The adhesion is correlated to surface tension as you mentioned. You can get ‘work of adhesion’ using ‘Owens-Wendt model’ [1].

[1] L.Q.N.Tran et al., Understanding the interfacial compatibility and adhesion of natural coir fibre thermoplastic composites, Compos Sci Technol 80, 23-30 (2013).

This reviewer was satisfied with the other responses. Thank you.

Response: Thanks for the reviewer’s insightful comments and suggestion to help us better understand the ‘adhesion’ and ‘interfacial compatibility’ more clearly, for an appropriate description in the manuscript. Following the reviewer’s suggestion, we measured the surface energy (γ) of the interlayers (PDIN, PDINO and PDINN) with the result provided in Supplementary Figure 23. Under the Owens-Wendt model, we also calculated interfacial energy (γ_{ca}) and the work of adhesion (W_a) for our interlayer lying on active layer component, with the results provided in Supplementary Table 7. It can be seen that whether PDINO or PDINN was contacted with the active layer component (PM6 or Y6), PDINN shows a lower γ_{ca} value. It is well known that the interfacial energy can be considered as an indicator of interfacial compatibility. Here, the lower γ_{sl} value means a high interfacial compatibility when PDINN was used as the interlayer.

As for the work of adhesion W_a , we also calculated its value based on those thermodynamics parameters according to the equation SI-6. But this calculation cannot include the factor contributed by the force of hydrogen bonding. More accuracy assessment of W_a can be obtained by directly using the pull-out tests. Without those tests, the discussion of W_a obtained by thermodynamics, have its limitation. So in the revised manuscript, the word

'adhesion' was changed to 'interfacial compatibility' based on the above discussion. The followings are the changes.

1) Change "However, the strong polar group could result in **poor adhesion of the interlayer** to the active layer, thus causing a physically poor contact in the devices." to "However, the strong polar group could result in poor physical contact of the interlayer to the active layer, thus causing carrier recombination in the devices.", in p. 2.

2) Change "good **adhesion** to the active layer of the OSCs via the hydrogen-bond interaction." revised to "good interfacial compatibility to the active layer of the OSCs via the hydrogen-bond interaction", in p. 2.

3) Change "provide **good adhesion** (chemical interaction) with the active layer below to suppress carrier recombination and interlayer decohesion" to "provide good interfacial compatibility with the active layer below to and suppress carrier recombination and interlayer decohesion.", in p. 4.

3) Change "PDINN shows a better **adhesion** on a wide range of active layers of OSCs compared with PDINO" to "PDINN shows a better interfacial compatibility on a wide range of active layers of OSCs compared with PDINO.", in p. 7.

4) Also, the claimant of "And the hydrogen bond of the CIMs can improve the mechanical adhesion between the active layer and CIMs in the OSCs.⁴¹" was removed. in p. 8.

Response to Reviewer #4:

The authors did a great job in taking into account most of the issues raised by the reviewers. Therefore, the manuscript may be suited for publication in Nature Communications provided that a remaining issue can be properly addressed. I am still not convinced by the description of the TA experiments and the conclusions drawn from these experiments for the following two reasons.

(1) Assignment of the transient absorption spectra: According to the newly added Figure S14, the transient absorption spectra consist of two spectral regions where the pump increases the transparency of the sample (at 600 nm – 670 nm and between 820 nm and 880 nm), while

there is a pump-induced decrease of the transmission in the remaining part of the spectrum. It is often proposed that an increased transmission is due to a ground state bleach (GSB), which is expected to show its maximum signal where the ground state absorption is largest. This may explain the strong positive signal at ca. 637 nm (where PM6 shows a strong absorption) but not of the second positive peak at 850 nm (because Y6 has its absorption maximum at 800 nm). On the other hand, a photoinduced decrease of transmission is by the absorption of newly created photogenerated species, such as excitons, polarons etc (excited state absorption, ESA). The text assigns the peak at 633 nm to a charge separated state and that at 770 nm to “the ESA” signal. What exactly is meant with charge separated state and why does it cause the peak at 637 nm. Why did the authors chose to plot the ESA at 780 nm, and why is the temporal decay of the ESA (apparently) different at e.g. 950 nm? The authors are asked to extend the discussion of the spectral features in the TA spectra and their specific decay properties.

Response: We agree with the reviewer that the signals of pump-induced transparency positive in the range of 600-670 nm and 820-880 nm are due to the ground state bleach (GSB). The signal at 630 nm can be safely assigned to the GSB of the donor (Fig. R1). Meanwhile, the signal at 850 nm is also induced by the GSB of the acceptor. The absorption peak of Y6 film locates near 840 nm (Fig. R1). The slightly red-shift of GSB feature of Y6 acceptor is likely caused by the ESA feature near 770 nm. The entanglement of such a broadband negative signal on the shorter-wavelength range leads to the red-shift of GSB feature of Y6 with respect to the absorption peak.

Indeed, the ESA features may be caused by different excited species including excitons, charge-transfer excitons and free polarons. For the charge-transfer excitons or the charge-separated states of free polarons, the electrons/holes are localized at the acceptor/donor sites respectively so that the GSB signals of both the acceptor and donor can be observed. The charge-separated states are much longer-lived if compared to the charge-transfer excitons. Following the well-established approach in literature (e.g., Baran et al., *Nat. Commun.* 9, 2059 (2018), Schlenker et al., *JACS* 134, 199661(2012), Lee et al., *Adv. Funct. Mater.* 20, 2945 (2010)), we measured the photo-induced absorption (PIA) spectrum to

identify the ESA feature of the charge-separated state. Upon weak continuous-wave excitation, the PIA is mainly contributed by the long-lived charge-separated state of free polarons. As shown in Fig. R2, the PIA feature of the charge-separated state can be observed in the bands centered at 780 nm and 950 nm. Nonetheless, the absorption cross sections for electron and hole polarons are possibly different, which may result in the slight difference in the dynamics probed at 770 nm and 980 nm. We have included the additional data and PIA spectrum in Supplementary Fig. 14 in the revised SI and the relevant explanation in the caption of Supplementary Fig. 14.

Figure R1. Absorption spectrum of PM6:Y6 film. Absorption peaks of PM6 and Y6 appears at 630 nm and 840 nm respectively.

Figure R2. The PIA spectrum recorded from a PM6:Y6 film. The CS state show two main absorption features around 770 nm and 980 nm.

(2) *Interpretation of the transient properties: The transient data in Figure 3b,c and in Figure S15 display a continuous decrease of the signal over several microseconds, and this decrease becomes accelerated at higher fluence. I agree with the authors that this long term decay is dominated by bimolecular recombination. But what is not clear to me is the sudden (early) decay of the transient signal in the sample with PDINO. The authors propose that this decay is due to trapping of charges at surface traps. I see two problems with this interpretation. First, if the signal at 633 nm stems from the ground state bleach on PM6, why would trapping of charges (by e.g. impurities and surface defects) restore the original PM6 absorption properties. Instead, trapping would slow down the return of the photogenerated electron back to the polymer backbone, thereby prolonging the signal decay. Second, the timescale of this early decay is 10-30 ns. If this decay would be due to charge carrier trapping at surface states, a considerable fraction of photogenerated electrons and/or holes must be able to travel to the surface. Since there is no electric field applied, this motion must be due to diffusion. I strongly doubt that the diffusion of electron and holes in the PM:Y6 blend is fast enough to allow them travelling to the surface within few tens of nanoseconds. Therefore, the authors need to carefully reconsider their interpretation.*

Response: We appreciate the reviewer's approval of our assignment of the long-term decay of the CS state as the bimolecular recombination. Indeed, the decay dynamics of the CS state are complicated and susceptible to many factors including the carrier density, trapping, and diffusion. For the charge separated state of free polarons, the electron/polaron polarons are localized at the acceptor/donor site so that the dynamics of free polarons may be reflected on the GSB signals, which explains why the trapping dynamics of charges is also captured in the bleach signal. Of course, the ESA signal represents directly the dynamics of the charge-separated states.

Carrier trapping may cause very different effects on the charge dynamics depending on the energetic landscape of the trapping centers. The presence of trapping centers like impurity or surface states provide additional relaxation channel of excited carriers, resulting in a faster decay of carriers as we proposed in the sample with PDINO. We agree with the reviewer that the signal decay may be prolonged if the carriers can escape from the trapping centers with

thermal activation. The charges undergoing such a trapping and de-trapping channel contribute to the photocharge generation which may not significantly reduce the IPCE. In contrast, the carriers trapped by deep level centers will recombine directly to the ground state prior to the de-trapping process which is a loss channel without contribution to charge generation. The data recorded from the sample with PDINO is consistent with the latter case with faster decay and reduced IPCE in comparison with the sample with PDINN. The density of surface trap states is possibly higher in the sample with PDINO.

The reviewer's concern on the charge diffusion is insightful. The charge mobility in the PM6:Y6 blend characterized by static measurements is on the level of $10^{-4} - 10^{-3} \text{ cm}^2/\text{Vs}$ (Yuan et al., *Joule* 3, 1140 (2019)). According to the Einstein relation of diffusion coefficient, the average diffusion distance at a delay time of 20 ns at room temperature can be estimated to be on the order of 10 nm with no voltage applied. The result suggests a small portion of charges may reach the surface as noted by the reviewer. While more in-depth study is required for a comprehensive understanding of the exact mechanism, the discrepancy is possibly due to the underestimation of the nonequilibrium charge mobility. As reported in literatures (Burke and Mc Gehee, *Adv. Mater.* 26, 1923 (2014), Pranculis et al., *JACS* 136, 11331(2014)), the local charge carrier mobility in OPV blends is highly time-dependent. On the timescale of 10 ns or shorter, the dynamic mobility, orders of magnitude larger than the late-stage value, should be much higher than the static value measured by time-integrated approaches. Considering the high nonequilibrium charge mobility, the portion of charges diffused to the surface is likely much higher than that considered statically above, which is possibly responsible for the faster decay component with amplitude ratio of $\sim 20\%$ in the trace recorded from the sample with PDINO. We have included the relevant discussion as the followings on p.15.

“The experimental details are described in the Supplementary Information. On the timescale of 10s of ns and longer, the TA data are mainly induced by the charge-separated state of free polarons as confirmed by the photo-induced-absorption measurements (Supplementary Figure 14). In the PM6:Y6 blend, the charge-separated state is configured with electron polaron at the acceptor site and hole polaron at the donor site, which may be

captured by the excited-state absorption (ESA) features of the free polarons and the ground-state bleach (GSB) features of donors and acceptors (Supplementary Figure 14). With increasing pump fluence, the decay of free polarons becomes faster due to bimolecular recombination process (Supplementary Figure 15). In comparison with the dynamics in the sample with PDINO, the charge recombination in the sample with PDINN is slower at the early stage (Fig. 3b and 3c), implying that the CIM of PDINN suppresses some loss channel. The different dynamics in the two samples are possibly related to the charge trapping due to the impurity and surface states. For the static transport measurements,¹⁰ the average diffusion distance at a delay time of 20 ns can be estimated to be on the order of 10 nm at room temperature with no voltage applied. The diffusion length may be much longer if the time-dependent local charge mobility is considered^{61,62}. The charges that diffuse to the surface may be trapped, resulting in a faster decay, which may reduce the IPCE if the charges recombine to the ground state prior to the thermally-activated de-trapping.”

REVIEWERS' COMMENTS:

Reviewer #1 (Remarks to the Author):

The points raised by this reviewer in the previous round of review have been properly addressed. The reviewer thus recommend the acceptance of this manuscript by Nature Communications without further change.

Reviewer #4 (Remarks to the Author):

The authors have further improved the quality of their manuscript by providing additional data and interpretations. I also appreciate the new Figure S14 including the results of cw photoinduced absorption measurements.

While these data are highly valuable, there is still an issue with the interpretation of the rapid first order decay of the transient signal in the PDINO-treated blend. Here, the authors still insist on their interpretation that this rapid decay of the TAS signals is due to charge carrier recombination through deep traps situated at the interface between the blend and the PDINO layer. To explain why surface traps can induce such fast decays despite the fairly low steady state diffusivities of electrons and blends in PM6:Y6, the authors point to the high mobility of photogenerated carriers at early time scales, thereby referring to early work by Burke and McGehee, and by Pranculis). In fact, there are more reports on high initial mobilities of photogenerated carriers, and some of those indeed suggested that electrons in PCBM-based blends can actually transfer through most of the active layer within 50 ns (e.g. 10.1002/adfm.201400404). However, these measurements were performed under a reverse bias of -1 V where the carrier motion is strongly enhanced by drift. More importantly, in order to recover the GSB of the PM6 (or Y6), it is not sufficient that one type of carrier gets trapped at a surface state. Instead, a photogenerated electron and a photogenerated hole both need to diffuse to the same surface state and recombine there. In my opinion, this scenario is very unlikely under the given conditions. Therefore, I still doubt that the observed rapid decay of the TAS signals is due to trapping and recombination at surface states.

Response to Reviewer #4:

The authors have further improved the quality of their manuscript by providing additional data and interpretations. I also appreciate the new Figure S14 including the results of cw photoinduced absorption measurements.

While these data are highly valuable, there is still an issue with the interpretation of the rapid first order decay of the transient signal in the PDINO-treated blend. Here, the authors still insist on their interpretation that this rapid decay of the TAS signals is due to charge carrier recombination through deep traps situated at the interface between the blend and the PDINO layer. To explain why surface traps can induce such fast decays despite the fairly low steady state diffusivities of electrons and blends in PM6:Y6, the authors point to the high mobility of photogenerated carriers at early time scales, thereby referring to early work by Burke and McGehee, and by Pranculis). In fact, there are more reports on high initial mobilities of photogenerated carriers, and some of those indeed suggested that electrons in PCBM-based blends can actually transfer through most of the active layer within 50 ns (e.g. 10.1002/adfm.201400404). However, these measurements were performed under a reverse bias of -1 V where the carrier motion is strongly enhanced by drift. More importantly, in order to recover the GSB of the PM6 (or Y6), it is not sufficient that one type of carrier gets trapped at a surface state. Instead, a photogenerated electron and a photogenerated hole both need to diffuse to the same surface state and recombine there. In my opinion, this scenario is very unlikely under the given conditions. Therefore, I still doubt that the observed rapid decay of the TAS signals is due to trapping and recombination at surface states.

Response: We are grateful that the reviewer appreciates the improvement made in the revised manuscript. Basically, the reviewer agrees with us that the transient mobility is orders of magnitude large than the value evaluated by static transport measurements as reported in the noted literature. That is, the diffusion length at the initial stage (< 50 ns) is significantly longer than the value (~ 10 nm) calculated with the static charge mobility. In the blend film of thickness of ~ 100 nm, a certain portion of photo-induced charges may be trapped by the surface states at the interface, which is responsible for the dynamics probed under weak excitation. Notably, such a trapping process for either electron or hole can be captured by the dynamics of GSB with signal amplitude half of that for the electron-hole recombination. The reviewer raised the concern that the charge mobilities in the quoted literatures were performed in the devices with applying voltage. It is worth noting that the charge mobility is principally independent of the applied voltage to a first-order approximation. We agree with the reviewer that the mobility in a practical device without voltage applied is possibly not the same as the

value with external voltage since the voltage may alter the interfacial energy landscape in the blend. Nevertheless, it is reasonable to expect the transient mobility with no voltage much higher than the static value. Moreover, a higher mobility by external voltage makes the effect of surface healing more important for device optimization. In addition to the surface effect, the decay dynamics may become faster if the transport of electrons and holes are imbalanced.

Given that direction evidence on surface trap needs further investigations, and relate investigation is out of scope of this manuscript, the discussion on the results were reorganized in the revised manuscript.